# Models of Heavy-Tailed Mechanistic Universality

Liam Hodgkinson [1]    Zhichao Wang [2][3]    Michael W. Mahoney [2][3][4]

## Abstract

Recent theoretical and empirical successes in deep learning, including the celebrated neural scaling laws, are punctuated by the observation that many objects of interest tend to exhibit some form of heavy-tailed or power law behavior. In particular, the prevalence of heavy-tailed spectral densities in Jacobians, Hessians, and weight matrices has led to the introduction of the concept of *heavy-tailed mechanistic universality* (HT-MU). Multiple lines of empirical evidence suggest a robust correlation between heavy-tailed metrics and model performance, indicating that HT-MU may be a fundamental aspect of deep learning efficacy. Here, we propose a general family of random matrix models—the high-temperature Marchenko-Pastur (HTMP) ensemble—to explore attributes that give rise to heavy-tailed behavior in trained neural networks. Under this model, spectral densities with power laws on (upper and lower) tails arise through a combination of three independent factors (complex correlation structures in the data; reduced temperatures during training; and reduced eigenvector entropy), appearing as an implicit bias in the model structure, and they can be controlled with an "eigenvalue repulsion" parameter. Implications of our model on other appearances of heavy tails, including neural scaling laws, optimizer trajectories, and the five-plus-one phases of neural network training, are discussed.

## 1. Introduction

Recent years have witnessed remarkable efforts directed toward providing a theoretical underpinning to "explain" the success of deep learning. Prominent examples include developments in overparameterized models (Jacot et al., 2018; Soltanolkotabi et al., 2018; Belkin et al., 2019; Nakkiran et al., 2021; Bartlett et al., 2021), nonvacuous generalization bounds (Bartlett et al., 2017; Dziugaite & Roy, 2017; Cao & Gu, 2019; Lotfi et al., 2022), statistical mechanics of learning (Martin & Mahoney, 2017; 2021b; Goldt et al., 2019; Bordelon et al., 2024a), random matrix theory (RMT) (Pennington & Worah, 2017; Liao et al., 2020; Mei & Montanari, 2022; Hu & Lu, 2022; Wei et al., 2022; Arous et al., 2024; Wang et al., 2024; Atanasov et al., 2025), and robust metrics to assess model quality (Jiang et al., 2020; Martin et al., 2021; Martin & Mahoney, 2021a). In general, however, most "predictions" for model performance provided by theory still tend to be *ad hoc*, providing limited practical utility beyond heuristics. As part of the effort to develop effective theoretical explanations, it is important to recognize, characterize, and explain empirical properties of deep learning models that are not shared by classical statistical models.

One of the more prominent and ubiquitous of such properties is the frequent appearance of *heavy-tailed distributions* for various objects of interest. This includes gradient norms (Simsekli et al., 2019) and loss curves (Hestness et al., 2017; Kaplan et al., 2020; Hoffmann et al., 2022). It also includes the singular values (eigenvalues of the product of a matrix with its conjugate transpose) of various matrices, including data covariance (Sorscher et al., 2022; Zhang et al., 2023), activation (conjugate kernel) (Pillaud-Vivien et al., 2018; Agrawal et al., 2022; Wang et al., 2023), Hessian (Xie et al., 2023), Jacobian (Wang et al., 2023), and weight matrices (Martin & Mahoney, 2021b). In classical settings, these objects typically exhibit (or are assumed to exhibit) Gaussian universality (Tao & Vu, 2011; Lu & Yau, 2022; Dandi et al., 2023). Indeed, much of classical statistical theory is built around the concentration associated with Gaussian universality. Our objective is to identify a new class of RMT models to describe heavy-tailed spectral behavior in each of these objects. Summarizing our contributions:

- **Modeling Framework.** We propose a *general modeling framework* for probabilistic analyses of trained neural network feature matrices, including the derivation of their spectral densities.

- **Beyond the Marchenko–Pastur Law.** We examine the *High-Temperature Marchenko-Pastur (HTMP) distribu-*

[1]School of Mathematics and Statistics, University of Melbourne, Australia [2]Department of Statistics, University of California, Berkeley CA, USA [3]International Computer Science Institute, Berkeley CA, USA [4]Lawrence Berkeley National Laboratory, Berkeley CA, USA. Correspondence to: Liam Hodgkinson <lhodgkinson@unimelb.edu.au>.

*Proceedings of the 42$^{nd}$ International Conference on Machine Learning*, Vancouver, Canada. PMLR 267, 2025. Copyright 2025 by the author(s).

*tion* (Dung & Duy, 2021), a recently-introduced RMT distribution that generalizes the Marchenko-Pastur (MP) law, and we propose its consideration in the origins of HT-MU. We argue that HTMP arises in the spectral density of *feature matrices with non-trivial structure*, influencing eigenvalue spacings according to a new parameter $\kappa$. As more structure is imposed, $\kappa$ decreases, resulting in heavier-tailed spectra. This phenomenon occurs *independently* of the behavior of matrix elements—the elements of HTMP models need *not* have heavy-tailed behavior.

- **Connections with Existing Heavy-Tailed Properties.** We apply the HTMP model to derive *neural scaling laws* (Kaplan et al., 2020), explain the mysterious appearance of *lower power laws* in optimizer trajectories (Hodgkinson et al., 2022), and explain the *five-plus-one phases of training* of Martin & Mahoney (2019).

We begin in Section 2 with a brief overview of current approaches to heavy-tailed spectral behavior and, in particular, heavy-tailed mechanistic universality (HT-MU). We introduce our modeling framework in Section 3, proposing that trained feature matrices can be modeled by a density of the inverse-Wishart type. Then, in Section 4, we derive the corresponding limiting spectral density, identifying three independent influences on heavy-tailed spectral behavior. Given these results, in Section 5, we apply our model to broader consequences of heavy tails in machine learning (ML). In Section 6, we provide a brief conclusion. Additional material, including proofs, may be found in appendices.

## 2. Heavy-Tailed Mechanistic Universality

*Heavy-tailed distributions* refer to a broad range of distributions (Resnick, 2007; Nair et al., 2022), most often with densities that decay slower than exponential, but the phrase is often (informally) used interchangeably with the prescription of *power laws*, which decay at a polynomial rate, $f(x) \sim cx^{-\alpha}$ as $x \to \infty$.[1] Importantly, as an empirical matter, many other densities also exist that can exhibit tails that are empirically indistinguishable from power laws (Clauset et al., 2009). These include log-normals (e.g., in layer norms (Hanin & Nica, 2020)) and exponentially-truncated power laws of the form $f(x) \sim cx^{-\alpha}e^{-\beta x}$ for small $\beta > 0$ (which can provide more informative fits for the spectra of weight matrices (Yang et al., 2023)).

In statistical physics (Sornette, 2006; Bouchaud & Potters, 2003) and the statistical physics of learning (Seung et al., 1992; Watkin et al., 1993; Engel & den Broeck, 2001), the presence of power laws is particularly special, suggesting an underlying universal mechanism driving their appearance.

Consequently, the frequency of heavy-tailed distributions appearing in modern ML suggests a new *Heavy-Tailed Mechanistic Universality* (HT-MU), a term coined in Martin & Mahoney (2020). Universality in this context describes systems near critical points, whose observables exhibit heavy-tailed, power-law behavior with shared exponents. We build our HT-MU models upon the work of Martin & Mahoney (2020); Martin et al. (2021); Martin & Mahoney (2021b), who identified heavy-tailed spectral distributions in pre-trained weight matrices, and posed the question: what constitutes universality in neural network weights? In RMT, universality denotes the emergence of system-independent properties derivable from a few global parameters defining an ensemble (Edelman & Rao, 2005; Edelman & Wang, 2013). In statistical physics, on the other hand, universality arises in systems with very strong correlations, at or near a critical point or phase transition; and it is characterized by measuring experimentally certain "observables" that display heavy-tailed behavior, with common—or universal—power law exponents. Although trained weight matrices are not truly random, but rather strongly correlated through training, RMT nonetheless provides a useful descriptive framework.

HT-MU has several important practical consequences. Most famously, spectral power laws in the activation matrices of deep neural networks give rise to *neural scaling laws* that are obeyed by the test loss with respect to both the number of parameters and training time (Bahri et al., 2024; Maloney et al., 2022). These laws reveal powerful selection criteria for designing compute-optimal models under a given budget (Kaplan et al., 2020; Hoffmann et al., 2022; Muennighoff et al., 2023; Paquette et al., 2024). In the absence of significant compute or the associated training/testing data, power laws encountered in the eigenspectra of weight matrices (and in the reciprocal of gradient norms; see Hodgkinson et al. (2022)) have been found to be unusually strong indicators of model quality. In particular, HT-MU underlies Heavy-Tailed Self-Regularization (HT-SR) Theory (Martin & Mahoney, 2021b), providing a framework for predicting trends in the quality of state-of-the-art neural network models—*without access to training or testing data* (Martin et al., 2021; Yang et al., 2023). Following these principles, practitioners can diagnose and improve—with very modest amounts of compute—underperforming models down to the level of individual layers (Zhou et al., 2023; Lu et al., 2024).

**Approaches to HT-MU.** Physical explanations for HT-MU range from the phase boundary of spin glasses, to directed percolation (Bouchaud & Potters, 2003), to self organized criticality (SOC) at the edge of chaos (Bertschinger et al., 2004; Cohen et al., 2021), to jamming transitions crossing from underparameterized to overparameterized models (Geiger et al., 2019; Spigler et al., 2019). It is challenging to translate these into a rigorous statistical model. On the other hand, RMT practitioners tend to examine the

---

[1] Power laws can also occur at the *bottom* part of the density, taking the form $f(x) \sim cx^{\alpha}$ as $x \to 0^+$, e.g., for gradient norms, as in Hodgkinson et al. (2022), and this will be relevant for us.

| Mechanism | Power Law Elements | Power Law Spectrum | Inverse Gamma |
|---|---|---|---|
| iid Heavy-Tailed Elements | ✓ | ✓ | × |
| Kesten Phenomenon | ✓ | ✓ | ✓/× |
| Population Covariance | ✓/× | ✓ | ✓/× |
| **Structured Matrices** (Thm. 4.1) | × | ✓ | ✓ |
| Empirical Observations (Features) | × | ✓ | ✓ |
| Empirical Observations (Weights) | × | ✓ | × |

*Table 1.* Comparison of various mechanisms, and their capacity to yield power laws, in feature matrix elements and feature matrix spectral densities, as well as their capacity to yield an inverse Gamma law for the spectral density in a neighborhood of zero. Empirical observations on feature matrices appear in Figure 1. Empirical observations on weight matrices appear in Martin et al. (2021); Martin & Mahoney (2021b).

interactions of individual models (Li & Sompolinsky, 2021; Wang et al., 2023), even under heavy-tailed data (Maloney et al., 2022; Bordelon et al., 2024b). Such analyses directly consider the effects of certain model architecture choices (e.g., depth) but they often do not apply to trained models, appear intractable for complex architectures, and obscure the underlying "reasons" for HT-MU we seek to identify.

To develop models of HT-MU, and to account for the broad range of objects in deep learning that exhibit heavy-tailed behavior, we focus on a class of *feature matrices*, encompassing activation, neural tangent kernel (NTK), and Hessian matrices. These are fundamental theoretical objects upon which other secondary ("observable") objects, including weight matrices, gradient norms, and loss curves, all depend. As in Martin & Mahoney (2021b), we use the machinery of RMT as a "stand-in" for a generative model of these feature matrices in state-of-the-art neural networks.

Our search for theories to underpin HT-MU revealed three primary categories of observable phenomena, which we outline and compare below and in Table 1. Most prominently, we consider the weight matrices of trained neural networks: such weight matrices are well-known to have spectral distributions that are strongly heavy-tailed, while having elements that are *not* heavy-tailed (Martin et al., 2021; Martin & Mahoney, 2021b). We also identify (in the right-most column of Table 1) a further universal property we observe in the spectra of NTK feature matrices: the left edge displays an inverse Gamma law.[2] See Figure 1 for a representative summary of the results. Additional details are provided in Appendix A.

- **Independent Heavy-Tailed Elements.** It is known that *independent* matrix elements exhibiting power laws with small tail exponents give rise to heavy-tailed spectral densities (Arous & Guionnet, 2008). This is less relevant for our discussion for two reasons: first, the elements of real

feature matrices are *not* independent (indeed, that was the original motivation for HT-SR theory and the introduction of HT-MU); and second, empirical results demonstrate that, while eigenvalues of weight matrices modern state-of-the-art models are heavy-tailed, elements are not (Martin et al., 2021; Martin & Mahoney, 2021b).

- **Kesten Phenomenon.** In natural systems, the ubiquity of power laws is often attributed to the SOC hypothesis, which asserts that dynamical systems in the neighborhood of a critical point exhibit power law behavior (Bak et al., 1987). In probability theory, this phenomenon arises from a mechanism discovered by Kesten (1973), where recursive systems on the edge of stability exhibit heavy-tailed stationary fluctuations. Considered with respect to gradient descent steps, Kesten's mechanism can explain the origins of heavy-tailed *size* fluctuations in the stochastic optimizer (Hodgkinson & Mahoney, 2021; Gurbuzbalaban et al., 2021); and analyses treating the neural network architecture itself as recursive reveal similar findings (Vladimirova et al., 2018; Hanin & Nica, 2020; Zavatone-Veth & Pehlevan, 2021). Here, too, though, theoretical results disagree with empirical results: it is the distribution of eigenvalues that is heavy-tailed, *not* the elements (Martin et al., 2021; Martin & Mahoney, 2021b). Indeed, the Kesten phenomenon in its currently-studied form primarily seems to occur for chaotic training behavior (Yang et al., 2023). It is an open question how to extend the results of Hodgkinson & Mahoney (2021); Gurbuzbalaban et al. (2021) to perform a Kesten-like iteration in the spectrum domain using free probability (which is of interest since it may reveal heavy tails in eigenvalues but not elements).

- **Heavy-Tailed Population Covariance.** A popular hypothesis is that the complex correlations exist in the data, are expressed as heavy-tailed spectra, and are passed onto features during training. This shifts the origin of HT-MU from the model to the data. Covariances of large datasets often exhibit power law spectra, and analyses centered around this approach have had predictive success for simple models (Paquette et al., 2024; Li et al., 2023). Correlations in the data clearly play a significant role, but implicit model biases should also play a role. Otherwise, different model architectures trained on the same dataset should exhibit similar power laws (see Section 4.3), which empirical results do *not* display (Martin et al., 2021; Yang et al., 2023).

**A New Approach using Structured Matrices.** We propose an alternative approach, one which considers the effect of implicit model bias exhibiting structure in the feature matrices. Our claims regarding the observed tail behavior of feature and weight matrices (as in Table 1) are summarized in the following metatheorem.

---

[2]The reciprocal of the exponentially truncated power law is the *inverse Gamma law*: $f(x) \sim cx^\alpha e^{-\beta/x}$ as $x \to 0^+$.

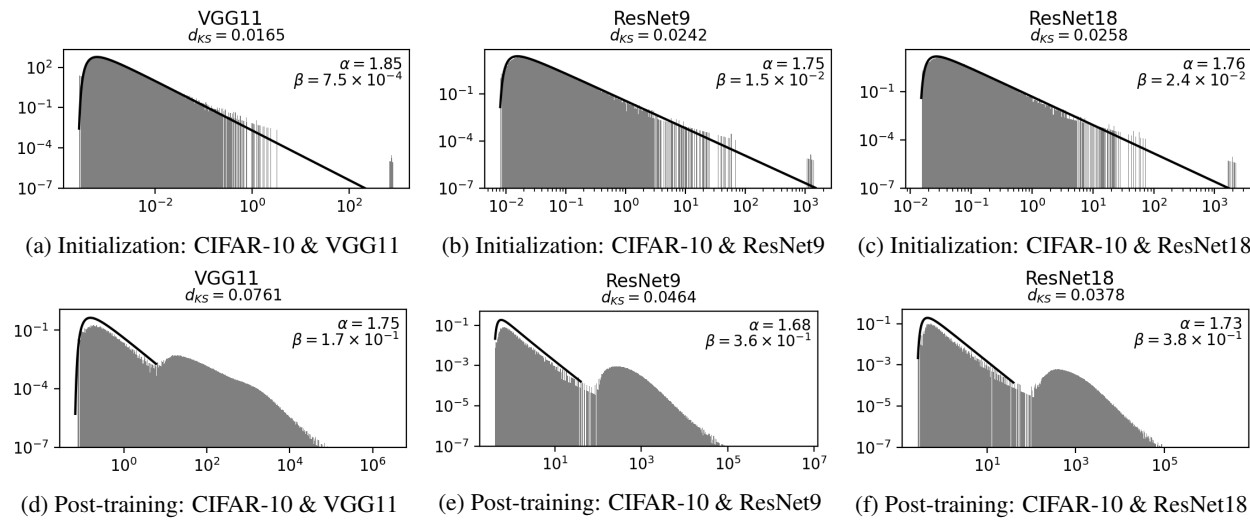

*Figure 1.* Distributions of spectral values (and inverse Gamma fits near zero) of the NTK matrix for `VGG11`, `ResNet9`, and `ResNet18` models trained on 1000 randomly-sampled datapoints from the CIFAR-10 dataset, at initialization and post-training.

---

> **Metatheorem** (Theorem 4.3 and Proposition 5.3)**.**
> - *Spectral densities of trained **feature matrices** can be modeled as the free multiplicative convolution of the label covariance spectrum and the reciprocal of an HTMP distribution, exhibiting both an inverse-Gamma law at 0 and a power law at $\infty$.*
> - *Spectral densities of trained **weight matrices** can be modeled with the HTMP distribution, exhibiting an exponentially-truncated power law at $\infty$.*

## 3. Modeling Framework

Tracking the precise spectral evolution of neural networks throughout training is generally challenging, both due to theoretical complexity and practical limitations (e.g., limited access to large-scale training dynamics). Instead, we propose a general modeling approach grounded in an entropic regularization perspective on stochastic optimization.

### 3.1. Entropic Regularization Setup

Let $\Theta$ be model coefficients with an initial density $\pi_\Theta$. To allow for possible early stopping in the optimization, we consider a stochastic optimizer that minimizes the loss by monotonically reducing the Kullback-Leibler divergence to a distribution of loss minimizers, representing "optimal" behavior of stochastic gradient descent (Chaudhari & Soatto, 2018). Define the stochastic minimization operation for any function $f(\Theta)$, $\mathrm{smin}_\Theta^{\pi_\Theta,\tau}$, with temperature $\tau > 0$ as:

$$\mathop{\mathrm{smin}}_{\Theta}^{\pi_\Theta,\tau} f(\Theta) := \min_{q\in\mathcal{P}}\left[\mathbb{E}_{q(\Theta)}f(\Theta) + \tau d_{\mathrm{KL}}(q\|\pi_\Theta)\right], \quad (1)$$

where $\mathcal{P}$ is the set of probability densities on the support of $\pi_\Theta$. We let $\mathrm{argsmin}_\Theta^{\pi_\Theta,\tau}$ denote the corresponding minimizing distribution $q(\Theta)$, provided it exists and is unique.

Stochastic optimization models of the form (1) have been considered previously (Mandt et al., 2016), and they have strong links to Bayesian inference (see Germain et al. (2016) and Appendix B), which itself has strong links to the statistical physics of generalization (Mezard & Montanari, 2009). In particular, applying (1) to the training loss optimizes a PAC-Bayes bound on the test error (see Appendix B). Equation (1) generalizes the approach of Xie et al. (2023), is equivalent to early-stopped graph heat-kernel diffusion (Mahoney & Orecchia, 2011), and (in ridge regression) coincides with known regularizers for early stopping with stochastic gradient descent (Sonthalia et al., 2024). As $\tau$ decreases during training, an optimizer following (1) smoothly interpolates between $\pi_\Theta$ and the final "optimal" density.

Since the features are of primary interest in our study, we split an arbitrary learning task into a subproblem of finding optimal model coefficients for a prescribed set of features, and a main problem of identifying those features. For example, as in Golub & Pereyra (1973), a neural network can be separated into its final layer and its final activations (representing trained features). Let $L$ be a loss function that depends on a collection of model coefficients $\Theta$ and features $\Phi$ with initial densities, $\pi_\Theta$ and $\pi_\Phi$, respectively, where $\pi_\Theta$ can depend conditionally on $\Phi$. In general, the optimally-trained features satisfy $\Phi^* = \mathrm{argmin}_\Phi \min_\Theta L(\Theta, \Phi)$, if a unique minimizer of $L$ in $\Theta$ exists for each $\Phi$. Replacing deterministic minimization in this expression with our stochastic optimizer (1), the probability density of "optimal features" $\Phi^*$ for $L(\Theta, \Phi)$ is computed in Proposition 3.1. We present its proof in Appendix C.1.

**Proposition 3.1** (OPTIMAL FEATURE DENSITY)**.** *Denote the minimizer of the feature density as $q(\Phi) := \mathrm{argsmin}_\Phi^{\pi_\Phi,\eta} \mathrm{smin}_\Theta^{\pi_\Theta,\tau} L(\Theta, \Phi)$, where* smin *is defined by*

(1) *and $\tau, \eta$ are the temperature parameters for $\Theta$ and $\Phi$, respectively. The probability density function $q(\Phi)$ satisfies*

$$q(\Phi) \propto \mathcal{Z}_\tau(\Phi)^{\tau/\eta} \pi_\Phi(\Phi),$$

*where $\mathcal{Z}_\tau(\Phi)$ is marginal (Gibbs) likelihood with prior $\pi_\Theta$:*

$$\mathcal{Z}_\tau(\Phi) = \mathbb{E}_{\Theta \sim \pi_\Theta} \exp\left(-L(\Theta, \Phi)/\tau\right).$$

Proposition 3.1 suggests that a stochastic optimizer concentrates on regions with high marginal likelihood (also called *model evidence*), with the degree of concentration determined by the ratio $\tau/\eta$. Importantly, in this result, feature learning (controlled by $\eta$) is allowed to occur at a different rate than coefficient learning (controlled by $\tau$).

### 3.2. Master Model

We now apply Proposition 3.1 to reveal densities for three popular types of feature matrices: activation matrices; NTK matrices; and Hessian matrices. Of particular interest are the late stages of training, where $\tau, \eta \to 0^+$, and we will also assume that $\tau/\eta \to \rho > 0$. In each case we present below, we observe that the trained feature matrix generally follows an *inverse-Wishart-type* density (Mardia et al., 2024, §3.8), of the form (2). Hence, we propose a Master Model Ansatz for the probability densities of trained feature matrices, with some parameters $\alpha, \beta > 0$ and initial density $\pi$:

> **Master Model Ansatz.** For feature matrices $M$,
> $$q(M) \propto (\det M)^{-\alpha} e^{-\beta \cdot \mathrm{tr}(\Sigma M^{-1})} \pi(M). \quad (2)$$

- **Activation Matrices.** Following Golub & Pereyra (1973); Pillaud-Vivien et al. (2018), consider a multilayer neural network with $m$ outputs, $f : \mathcal{X} \to \mathcal{Y} \subset \mathbb{R}^m$, which is parameterized in terms of its final linear layer, $W = (w_{jk}) \in \mathbb{R}^{d \times m}$, and a feature vector $\varphi : \mathcal{X} \to \mathbb{R}^d$ of the last hidden layer. That is, the $k$-th output of $f$ is defined by $f_k(x) = \sum_{j=1}^d w_{jk} \varphi_j(x)$ for $k \in [m]$. For a dataset $\mathcal{D} = \{(x_i, y_i)\}_{i=1}^n \subset \mathcal{X} \times \mathcal{Y}$, we can investigate the ridge regression problem of minimizing $L(W, \Phi) = \|\Phi W - Y\|_F^2 + \mu \|W\|_F^2$, where $\mu > 0$, $\Phi_{ij} = \varphi_j(x_i)$ and $Y = (y_i)_{i=1}^n \in \mathbb{R}^{n \times m}$. For $\pi_\Theta = \mathcal{N}(0, \sigma^2 I)$ and $\tilde{\sigma}^2 = \frac{\sigma^2}{1 + \frac{2\mu\sigma^2}{\tau}}$, the marginal likelihood takes the form

$$\mathcal{Z}_\tau(\Phi) \propto \frac{\exp\left(-\frac{1}{2}\mathrm{tr}(Y^\top (\tilde{\sigma}^2 \Phi\Phi^\top + \frac{\tau}{2}I)^{-1} Y)\right)}{\det(\tilde{\sigma}^2 \Phi\Phi^\top + \frac{\tau}{2}I)^{m/2}}. \quad (3)$$

See Appendix C.2 for a derivation of (3). We can now apply Proposition 3.1 to this model to find the minimizing density for $\Phi$. Let $\Sigma = YY^\top$, and consider a change of variables to $M = (1 + \frac{2\mu\sigma^2}{\tau})^{-1} \Phi\Phi^\top + \frac{\tau}{2\sigma^2} I$. Then, given the density of $M$ before training as $\pi$, we can conclude that the density of $M$ after training is

$$q(M) \propto (\det M)^{-\rho m/2} \exp(-\tfrac{1}{2}\rho\sigma^2 \mathrm{tr}(\Sigma M^{-1})) \pi(M).$$

This is in the form of the Master Model Ansatz, with $\alpha = \rho m/2$ and $\beta = \rho\sigma^2/2$.

- **Neural Tangent Kernel (NTK).** Consider the NTK Gram matrix $J(\Phi) \in \mathbb{R}^{mn \times mn}$, where each $m \times m$ block is given by $J(\Phi)_{ij} = Df_{\Theta,\Phi}(x_i)^\top Df_{\Theta,\Phi}(x_j)$, where $Df_{\Theta,\Phi} \in \mathbb{R}^{d \times m}$ is the Jacobian of the model $f_{\Theta,\Phi} : \mathcal{X} \to \mathcal{Y} \subset \mathbb{R}^m$ with parameters $\Theta \in \mathbb{R}^d$. Coined by Jacot et al. (2018), the NTK is central to the analyses of generalization performance in neural networks (Huang & Yau, 2020). The NTK approximation, treated in Rudner et al. (2023) and Wilson et al. (2025), is the linearized model $f_{\Theta,\Phi}(x) \approx f_{\Theta^*,\Phi}(x) + Df_{\Theta^*,\Phi}(x) (\Theta - \Theta^*)$, under the loss

$$L(\Theta, \Phi) = \|f_{\Theta,\Phi}(X) - Y\|_F^2. \quad (4)$$

By the same arguments used to derive (3), if $\bar{Y} = \mathrm{vec}(Y - f_{\Theta^*,\Phi}(X)) \in \mathbb{R}^{mn}$, then

$$\mathcal{Z}_\tau(\Phi) \propto \frac{\exp(-\frac{1}{2}\mathrm{tr}(\bar{Y}^\top (\sigma^2 J(\Phi) + \frac{\tau}{2}I)^{-1} \bar{Y}))}{\det(\sigma^2 J(\Phi) + \frac{\tau}{2}I)}.$$

Applying Proposition 3.1 for $M = J(\Phi)$, we obtain

$$q(M) \propto (\det M)^{-\rho/2} \exp\left(-\frac{\rho\sigma^2}{2}\mathrm{tr}(\Sigma M^{-1})\right) \pi(M), \quad (5)$$

which is also in the form of the Master Model Ansatz, with $\alpha = \rho/2$ and $\beta = \rho\sigma^2/2$. Given that the Fisher information matrix possesses the same nonzero eigenvalues as the corresponding NTK, we can derive a comparable result for the Fisher information matrix as well; we omit the details for brevity. The NTK approximation is known to be effective for models in the late stages of training (Fort et al., 2020). A question is: can we justify (5) without appealing to a linear approximation? To answer this affirmatively, consider the setting where $d > mn$, where Hodgkinson et al. (2023) have developed asymptotic approximations to the marginal likelihood under the loss (4) for very general models in the interpolating regime. Their arguments lead to a similar finding to (5), when the model is trained to implicitly regularize a lower bound on its variance. See Appendix D for details.

- **Hessian Matrix.** For more general losses $L(\Theta, \Phi)$, computing the marginal likelihood becomes intractable, but it can be well-estimated by Laplace approximation in the regime where $\tau \to 0^+$. Letting $\Phi$ denote appropriate hyperparameters or architecture choices, Simon (2015, Theorem 15.2.2) immediately gives

$$\mathcal{Z}_\tau \propto \det(\nabla_\Theta^2 L(\Theta^*, \Phi))^{-1/2} e^{-\frac{1}{\tau} L(\Theta^*, \Phi)} [1 + \mathcal{O}(\tau)].$$

Applying Proposition 3.1, up to an $\mathcal{O}(\tau)$-error term,

$$q(\Phi) \propto \det(\nabla_\Theta^2 L(\Theta^*, \Phi))^{-\rho/2} e^{-\frac{1}{\eta} L(\Theta^*, \Phi)} \pi_\Phi(\Phi).$$

This expression need *not* satisfy the Master Model Ansatz. However, for losses of the form (D.1) (see Appendix D), $\nabla_\Theta^2 L(\Theta^*, \Phi) = \sum_{i=1}^n Df(x_i)Df(x_i)^\top$ when $L(\Theta, \Phi) = 0$, so the spectrum of the Hessian is equivalent (up to zeros) to that of the NTK. Thus, the version of the Master Model Ansatz (5) (see Appendix D) also applies for the Hessian for small training loss.

# 4. Heavy-Tailed Spectral Behavior

With the Master Model Ansatz (2) for trained feature matrices in hand, we can explore potential theoretical explanations for HT-MU that fit within this structure. We begin by outlining a proposed family of random matrices that satisfies (2) and that exhibits spectral behavior (from Table 1) reminiscent of observations of HT-MU.

## 4.1. The HTMP Spectral Density

Obtaining a spectral density from the Master Model Ansatz (2) requires diagonalization: for $M$ a symmetric matrix, we can perform the change of variables $M \mapsto Q\Lambda Q^\top$, where $Q$ is an orthogonal matrix of eigenvectors and $\Lambda$ is a diagonal matrix of eigenvalues. The covariance matrix $\Sigma$ can be removed from (2) by a change of variables, so for now let $\Sigma = I$ (we return to the general $\Sigma$ scenario later in Section 4.3). With the effect of $\Sigma$ removed, only the influence of the density $\pi$ of feature matrices $M$ at initialization remains to be characterized, to completely determine the spectral density from (2).

Although the joint density of eigenvalues can be complicated, depending on $\pi$, we argue (see Section 4.2) that much of the behavior is captured by the extent of the eigenvalue repulsions. To isolate this effect, we consider the family of beta-ensembles (Dumitriu & Edelman, 2002), parameterized by $0 \leq \kappa \leq N$, for the joint density of eigenvalues:

$$q_\kappa(\lambda_1, \ldots, \lambda_N) \propto \prod_{i=1}^N e^{-V(\lambda_i)} \prod_{\substack{i,j=1,\ldots,N \\ i<j}} |\lambda_i - \lambda_j|^{\kappa/N}. \quad (6)$$

For (6) to follow (2), we must take $V(\lambda) = \lambda^{-\alpha}e^{-\beta\lambda^{-1}}$. Beta-ensembles are well-studied objects in RMT, but they are typically considered theoretical curiosities, rather than "physical" models of matrices (Dumitriu & Edelman, 2002; Forrester, 2010). The $1/N$ "high temperature" scaling has also been examined (Forrester & Mazzuca, 2021), but without application. We argue (in Section 4.2) that $\kappa$ determines the "degree of randomness" in the matrix model, or conversely, the rigidity of the matrix structure.

Fortunately, Dung & Duy (2021) have derived the spectral density for high-temperature beta-ensembles (6). We use their result to introduce a novel RMT class of *high-temperature Marchenko-Pastur* (HTMP) densities, parameterized by an aspect ratio $\gamma$ and a structure parameter $\kappa$, in Theorem 4.1. The HTMP class of densities is defined

explicitly and examined in greater detail in Appendix E. The proof of this theorem is deferred to Appendix E.2.

**Theorem 4.1** (HTMP). *Consider a sequence of matrices $M_N$ obeying the high-temperature inverse-Wishart ensemble* (6) *with $\kappa = \kappa(N)$. Assume $\gamma(N) := \frac{\kappa/2}{\alpha - \kappa/2 - 1} \to \gamma$ for some constant $\gamma \in (0,1)$ as $N \to \infty$. The empirical spectral distributions of $\frac{2\gamma(N)\beta}{\kappa}M_N^{-1}$ converge to:*

(a) $\mathbf{MP}_\gamma$ *(Marchenko-Pastur) if $\kappa(N) \to \infty$; or*

(b) $\mathbf{HTMP}_{\gamma,\kappa}$ *(high-temperature Marchenko-Pastur) if $\kappa(N) \to \kappa$.*

HTMP densities generalize the celebrated MP law, with an additional shape parameter, $\kappa$, that permits interpolation between heavy-tailed spectra ($\kappa \to 0^+$) and MP spectra ($\kappa \to \infty$). We highlight the visual differences between MP and HTMP densities in Figure 2, with the key discrepancy lying in the tail behavior: HTMP spectral densities exhibit heavier tail behavior and have infinite support (even though the elements of HTMP need *not* have heavy-tailed behavior).

## 4.2. Structured Feature Matrices

Our next objective is to investigate the nature of the parameter $\kappa$ in the HTMP distribution and motivate the use of the high-temperature beta-ensemble (6) in the context of the Master Model (2). To do so, we shall first consider the joint density of eigenvalues when $\pi$ is uniform over different structured matrix classes. Our motivating feature matrix is the NTK matrix, defined as an $N \times N$ matrix comprised of $n \times n$ blocks, each of size $m \times m$, so $N = mn$. We consider cases where $\pi$ is uniform over: (a) Diagonal matrices; (b) Commuting block-diagonal matrices, where each $m \times m$ block on the diagonal commutes with each other, and all other blocks are zero; (c) Symmetric block-diagonal matrices, where each $m \times m$ block on the diagonal is an arbitrary symmetric positive-definite matrix, and all other blocks are zero; (d) Kronecker-like matrices, where the eigenvector matrix $Q$ is a Kronecker product $Q_1 \otimes Q_2$, where $Q_1 \in \mathbb{R}^{m \times m}$ and $Q_2 \in \mathbb{R}^{n \times n}$; and (e) arbitrary (symmetric) matrices. Activation and NTK matrices for neural networks have been hypothesized to exhibit Kronecker-like structures, as in case (d) (Martens & Grosse, 2015). In Appendix F, we derive the joint eigenvalue density for each of these classes.

The key observation, noted in Remark F.6, is that the joint densities for (a)–(e) all exhibit absolute differences of eigenvalues, $|\lambda_i - \lambda_j|$, differing mainly in how many such terms are multiplied together. This variation reflects differing degrees of eigenvalue repulsion, a consequence of the reduced randomness in the eigenvectors, as formalized in the *eigenvector-eigenvalue identity* (Denton et al., 2022, Theorem 1). Because structured matrices lack uniformly random eigenvectors, their joint eigenvalue densities are not symmetric, complicating the determination of a limiting spectral

| Structure | $\kappa^*$ |
|---|---|
| (a) Diagonal | $0$ |
| (b) Commuting block diagonal | $\frac{m}{n} - \frac{1}{2n}$ |
| (c) Symmetric block diagonal | $(m-1) \cdot \frac{mn}{mn-1}$ |
| (d) Kronecker-like matrix | $\frac{n}{m} + \frac{m}{n}$ |
| (e) Symmetric (no structure) | $mn$ |

*Table 2.* Behavior of $\kappa^*$ across matrix structures for matrices with $n \times n$ blocks of size $m \times m$. Assuming $n > m$, the first term in each expression provides the dominant behavior.

density. To make these densities symmetric, one can approximate them by symmetric polynomials; isolating the leading-order behavior leads to (6). Therefore, by proposing (6) as a variational family of approximations to the spectral density, for an arbitrary joint density of eigenvalues $q$, the temperature $\kappa^*$ can be identified by

$$\kappa^* = \underset{\kappa}{\arg\min}\, d_{\mathrm{KL}}(q_\kappa \| q). \qquad (7)$$

We consider the behavior of $\kappa^*$ for each of the five different matrix structures described above. See Appendix G for details. In particular, in Proposition G.1, we prove that $\kappa^*$ counts the number of eigenvalue repulsions in $q$, when it exhibits a simple product form, as in cases (a), (c), (e). The more complex structures, (b) and (d), require numerical methods. In Algorithm 1 of Appendix G, an efficient numerical procedure for estimating $\kappa^*$ is provided (satisfying a central limit theorem, see Proposition G.5). Using symbolic regression, the relationship between $\kappa^*$ and $m, n$ can be ascertained. Our findings are summarized in Table 2. We observe a direct correlation between the size of $\kappa^*$ and how restrictive the matrix structures are, with the most restrictive (diagonal) and least restrictive (arbitrary) cases occupying the smallest and largest possible values of $\kappa^*$.

### 4.3. Including Population Covariance

The most popular hypothesis behind the origin of heavy-tailed spectral behavior is the occurrence of this behavior in the data.[3] We can understand this in terms of the Master Model Ansatz as follows. For a fixed prior $\pi$, let $M_\Sigma$ be distributed according to the (6). Assuming a homogeneous prior, $M_\Sigma$, is equivalent in distribution to $\Sigma^{1/2} M_I \Sigma^{1/2}$. Thus, the spectral density of $M_\Sigma$ can be computed from $M_I$ and $\Sigma$ using Voiculescu's $S$-transform (Voiculescu, 1987). We then have the following proposition, which highlights how heavy-tailed spectra in $M_\Sigma$ can arise from heavy-tailed spectra in the population covariance $\Sigma$, assuming that the feature matrix $M_I$ for isotropic data exhibits *lighter-tailed* spectra than the data. This is true whenever $M_\Sigma = \Sigma^{1/2} M_I \Sigma^{1/2}$, regardless of whether (6) holds.

**Proposition 4.2.** *Assume that the spectral measure $\mu_\Sigma$ of $\Sigma$ satisfies $\mu_\Sigma((z, +\infty)) \sim z^{-\alpha} L(z)$ as $z \to \infty$, for a slowly*

---

[3]Even if this is less interesting from the perspective of HT-MU and this paper, we include this for completeness.

*varying function $L(z)$, and $\mathbb{E}\mathrm{tr}(M_I^{\alpha+1}) < +\infty$. Then the spectral measure $\mu_{M_\Sigma}$ of $M_\Sigma$ satisfies $\mu_{M_\Sigma}((z, +\infty)) \sim c_\alpha z^{-\alpha} L(z)$ as $z \to \infty$ for a constant $c_\alpha$.*

Proposition 4.2 follows from Kołodziejek & Szpojankowski (2022, Lemma 7.2). A similar argument shows that population covariance can also generate power laws for $M_\Sigma^{-1}$.

While spectral distributions of many datasets, including CIFAR-10 and large language datasets, tend to exhibit power law behavior (Clauset et al., 2009; Zhang et al., 2023), Proposition 4.2 "predicts" that if the population covariance is the *only* mechanism by which heavy-tailed spectra occur, then the tail exponents should not differ between model architectures. This is an example of the *power-law in, power-law out* (PIPO) principle. More recent analyses have shown how other decisions, including architectural decisions, can alter the power law (Maloney et al., 2022). However, these results hold only for specialized models. One advantage of our approach is that the direct influence of the data can be separated, so our conclusions depend only on the resulting structure of the feature matrix, independently of the data.

### 4.4. Tail Behavior

We now put everything together to state our main theorem.

**Theorem 4.3.** *Let $M_N$ denote the $N \times N$ Gram matrix of a trained feature matrix obeying the Master Model (2). For $\rho_N$, the empirical spectral distribution of $M_N$, and $\Sigma$, the label covariance matrix with spectral measure $\mu_\Sigma$, we have:*

$$\rho_N(\lambda) \to (\mu_\Sigma \boxtimes \rho)(\lambda) \quad \textit{as } N \to \infty, \qquad (8)$$

*where $\boxtimes$ denotes free multiplicative convolution. Under the ensemble (6), the limiting density $\rho(\lambda) = \lambda^{-2} \rho_{\mathrm{HTMP}}(\lambda^{-1})$ if $\kappa$ is finite; $\rho(\lambda) = \lambda^{-2} \rho_{\mathrm{MP}}(\lambda^{-1})$ if $\kappa = \infty$, where $\rho_{\mathrm{MP}}$ and $\rho_{\mathrm{HTMP}}$ are the probability density functions of $\mathbf{MP}_\gamma$ and $\mathbf{HTMP}_\gamma$ defined in Theorems E.1 and E.2, respectively. Additionally,*

- ***Power Law at $\infty$:** For some constant $c_+ > 0$,*

$$\rho(x) \sim \begin{cases} \textit{bounded support} & \textit{if } \kappa = \infty, \gamma \neq 1 \\ c_+ x^{-3/2} & \textit{if } \kappa = \infty, \gamma = 1 \quad \textit{as } x \to \infty. \\ c_+ x^{-\frac{\kappa}{2\gamma} - 1 + \frac{\kappa}{2}} & \textit{otherwise,} \end{cases}$$

- ***Inverse Gamma Law at $0$:** If $\kappa = \infty$, the support of $\rho$ is bounded away from zero; otherwise, for some $c_-, \beta_- > 0$,*

$$\rho(x) \sim c_- x^{-\frac{\kappa}{2\gamma} - 1 - \frac{\kappa}{2}} \exp\left(-\frac{\beta_-}{x}\right) \quad \textit{as } x \to 0^+.$$

*Remark 4.4.* The power law for the limiting density $\rho$ contains a tail exponent that *gets heavier as $\kappa$ decreases*, i.e., as the structure of the underlying matrix becomes more rigid. Interpreting this as increasing implicit model bias, our findings are in line with conjectures of Martin & Mahoney (2021b) and Simsekli et al. (2019), claiming that heavier tails imply stronger model biases, and therefore better model quality and generalization ability. Importantly, the elements of these models need *not* have heavy-tailed behavior.

To the best of our knowledge, our HTMP model represents the first RMT ensemble that captures key empirical properties of (strongly-correlated) modern state-of-the-art neural networks (Martin & Mahoney, 2021b; Martin et al., 2021).

## 5. Applications

Here, we describe how our theory relates to several heavy-tailed results observed empirically for neural networks.

### 5.1. Neural Scaling Laws

Perhaps the most famous appearances of power laws in deep learning are the *neural scaling laws*, which assert that the test loss at the end of training scales as a power law with respect to both the number of parameters, $d$, and the size of the dataset, $N$ (Hestness et al., 2017; Kaplan et al., 2020). In its modern form given by Hoffmann et al. (2022), using our notation, the test loss is observed to behave as $L \sim L_0 + \frac{A}{d^\alpha} + \frac{B}{N^\beta}$, for some constants $\alpha, \beta, A, B, L_0 > 0$. Since the total cost $C$ of training satisfies $C \propto Nd$, for a fixed computational budget, one can identify the optimal choices of $N$ and $d$ to achieve minimal test loss.

We present a neural scaling law for ridge regression on the activation matrix satisfying the spectral density function in Theorem 4.1. Unlike previous scaling law work, instead of assuming a power law in the dataset (e.g., as done by Wei et al. (2022); Defilippis et al. (2024); Paquette et al. (2024); Lin et al. (2024)), we show how the scaling limit depends on the power law in the feature matrix $\Phi$, discussed in Section 3. The following proposition is proven in Appendix C.3.

**Proposition 5.1** (NEURAL SCALING LAW). *Consider the activation matrix scenario in Section 3 with $m = 1$ and $\varphi : \mathcal{X} \to \mathbb{R}^d$. Suppose that each label $y_i = w_*^\top \varphi(x_i), i \in [n]$ and $\Phi$ satisfies the conditions of Theorem 4.1, with $\Sigma = I$ and parameters $\kappa$ and $\gamma$. For the solution $\hat{w} = \mathrm{argmin}_w L(w, \Phi)$, consider the generalization error*

$$L := \mathbb{E}_{x, w_* \sim \mathcal{N}(0, \frac{1}{d}I)}[(\varphi(x)^\top \hat{w} - y)^2]. \quad (9)$$

*Then, for $\mu = n^{-\ell}$ with $\ell \in (0,1)$, with high probability,*

$$L \asymp n^{-\ell(2 + \frac{\kappa}{2\gamma} - \frac{\kappa}{2})}, \quad as\ n \to \infty.$$

We compare this result with previous work in Appendix H. Several works have outlined theories to explain the appearance of neural scaling laws (Bahri et al., 2024; Maloney et al., 2022). Each assumes the eigenvalues of a covariance matrix (either the final layer of activations or $J$) exhibit power law decay $\lambda_k \sim ck^{-s}$. We remark that the shape of the spectrum of $J$ *after training* is key, as the spectrum at initialization cannot predict neural scaling laws (Vyas et al., 2023; Bordelon et al., 2024b) for feature learning.

Similar to neural scaling laws is the appearance of power law decay in the rate of convergence to zero in the training loss. That is, letting $f_T$ denote the model obtained at epoch $T$, then $\mathcal{L}_N(f_T) \sim cT^{-\alpha}$ as $T \to \infty$ (Agrawal et al., 2022). Velikanov & Yarotsky (2024) provide a comprehensive theoretical framework establishing power law rates of convergence across multiple optimizers (see also Velikanov & Yarotsky (2021)), under the assumption that the spectral density of $J$ satisfies a power law near zero.

### 5.2. Optimizer Trajectories

Contrary to the popular belief that stochastic optimizers exhibit Gaussian fluctuations, Mandt et al. (2016); Simsekli et al. (2019) observed that the distribution of gradient norms of large neural networks during training exhibits a power law. This behavior manifests in heavy-tailed fluctuations during training, and it enables rapid escape from basins of poor-performing models in the loss landscape (Nguyen et al., 2019). We use the terms *lower* and *upper power law* to correspond to polynomial behavior in the distribution of gradient norms about zero and infinity, respectively:

$$\mathbb{P}(\|\hat{\nabla} L_N\| \leq x) \sim C_- x^\alpha \quad as\ x \to 0^+, \quad (10a)$$

$$\mathbb{P}(\|\hat{\nabla} L_N\| > x) \sim C_+ x^{-\beta} \quad as\ x \to \infty. \quad (10b)$$

Both behaviors have been observed in practice, with generalization performance tied to the lower tail exponent $\alpha$ (Hodgkinson et al., 2022). There has been significant theoretical justification for the upper power law (10b) in terms of the Kesten mechanism (Hodgkinson & Mahoney, 2021; Gurbuzbalaban et al., 2021; 2022), but there has been little justification for the lower power law (10a). Our analysis can partially explain this too. The following proposition is proven in Appendix C.4[4].

**Proposition 5.2.** *Consider the loss function (4). Assuming the residuals $f_{\Theta, \Phi}(X) - Y$ are normally distributed and independent of an inverse Wishart-distributed NTK matrix $J$, the stochastic gradients $\hat{\nabla} L_N$ satisfy (10).*

### 5.3. The 5+1 Phases of Training

The more difficult power laws to explain are those appearing in the weight matrices themselves (Martin & Mahoney, 2019; 2020; 2021b). Power law exponents in the spectrum of weight matrices are simple to compute and are strongly predictive of model performance (Martin et al., 2021; Yang et al., 2023; Zhou et al., 2023). In a sequence of papers, Martin & Mahoney observed six classes of behaviors in trained weight matrices, with a smooth transition from a random-like MP law to a heavy-tailed density, before experiencing a "rank collapse." Excluding the rank collapse phase, the five primary phases are: (a) Random-Like: pure noise, modeled by a MP density; (b) Bleeding-Out: some spikes occurring outside the bulk of density; (c) Bulk+Spikes: spikes are distinct and separate from the MP bulk; (d) Bulk-Decay: tails extend so that the support of the density is no longer finite;

---

[4]Proposition 5.2 for the beta-ensemble (6) remains an open problem, although we conjecture that it holds universally

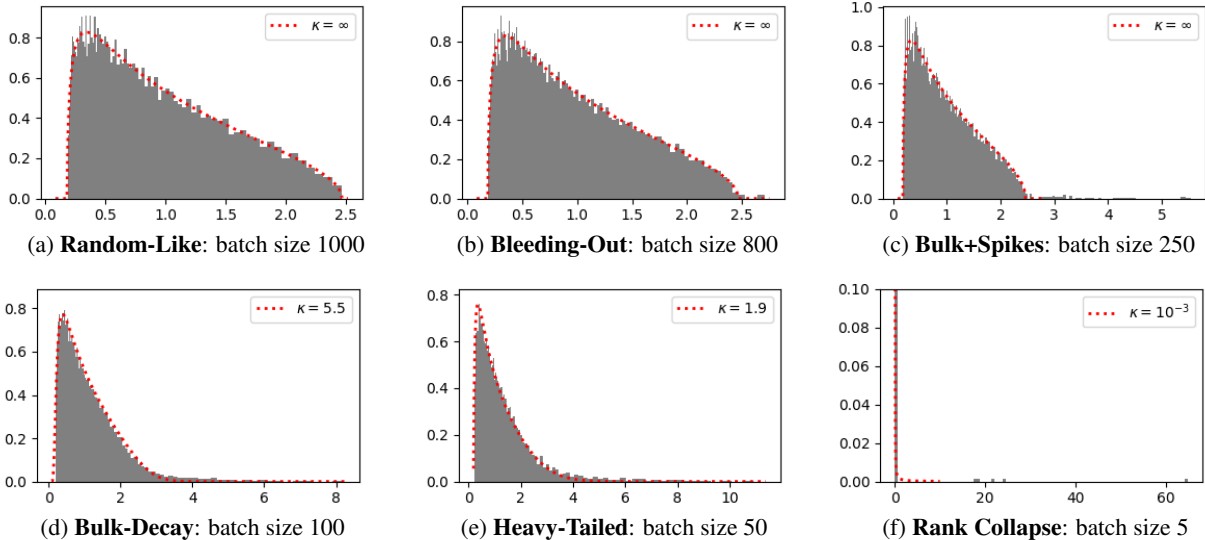

*Figure 2.* The "5+1 phases of training" in weight matrices, as estimated by Theorem 4.1. Compare with Figure 12 of Martin & Mahoney (2021b) (which is Figure 14 in the technical report version of their paper). All spectral densities (black) are compared to a corresponding MP density with an aspect ratio $\gamma = 0.3255$. The red dashed lines are density functions in Theorem 4.1 with different $\kappa$. The top row comprise cases with $\kappa = \infty$; the last row involves $\kappa = 5.5$, $\kappa = 1.9$ and $\kappa = 10^{-3}$ from left to right. See Section 5.3 for details.

and (e) Heavy-Tailed: the tails become more heavy-tailed, exhibiting the behavior of a (possibly truncated) power law. The transition from (a) to (e) is also seen in Thamm et al. (2022), as well as some non-uniformity in the eigenvectors, indicative of matrix structure. This smooth transition between multiple phases is a primary motivation of this work. We find that this behavior is displayed by a combination of a nontrivial covariance matrix to capture the spikes and the HTMP class with decreasing $\kappa$. Indeed, we have the following, Proposition 5.3, proved in Appendix C.5.

**Proposition 5.3** (WEIGHT MATRICES). *Consider the activation matrix scenario in Section 3. Let $A = W^\top W$, where $W = \arg\min_W L(W, \Phi)$. Assume $M = \Phi\Phi^\top$ satisfies (2) and $\pi(\Sigma M)$ has joint eigenvalue density (6) with $V \equiv 1$. As $m, d \to \infty$, for $\gamma$ in Theorem 4.1, the empirical spectral distribution of $\frac{\beta}{\alpha - \frac{\kappa}{2} - 1} A$ converges to $\mathbf{HTMP}_{\gamma,\kappa}$.*

In Figure 2, we show that the HTMP family, $\mathbf{HTMP}_{\gamma,\kappa}$, provides good fits to observed spectral distributions across the five phases of training, compared with Martin & Mahoney (2021b, Figure 2). In Figure 2, we plot the eigenvalue distributions of the fully connected layer of MiniAlexNet (a smaller AlexNet used by Zhang et al. (2021)) after training, using different batch sizes. In all the cases, we ensure the training losses are smaller than 0.01. Figure 2 indicates how training dynamics affect the heaviness of the tail, related to $\kappa$, for the trained weight matrices.

## 6. Conclusions

Motivated by complementary lines of work describing heavy-tailed properties in state-of-the-art neural networks

structure and/or dynamics, we present a modeling framework for spectral densities of trained feature matrices, including activation matrices, Hessian matrices, and NTK matrices, culminating in a Master Model Ansatz (2). Based on this, we explored several factors contributing to the origins of HT-MU. In line with the PIPO hypothesis, if the covariance of the data exhibits heavy-tailed spectra, trained feature matrices should be expected to adopt this behavior. Moreover, if a model exhibits bias on the distribution of eigenvectors, reducing eigenvector entropy, the joint density of eigenvalues can be approximated by a high-temperature inverse-Wishart ensemble. The local eigenvalue spacings can be characterized with a new parameter, $\kappa$, and we propose the use of the $\mathbf{HTMP}_{\gamma,\kappa}$ model for the limiting spectral density. This distribution becomes more heavy-tailed as $\kappa$ decreases (more implicit bias, see Theorem 4.1). This model was used to explain recent examples of heavy-tailed phenomena: obtaining new scaling laws; explaining the presence of lower power laws in optimizer trajectories; and describing the five phases of training from Martin & Mahoney (2019).

We anticipate that the HTMP density can serve as a valuable starting point for investigating heavy-tailed spectral phenomena. Nevertheless, we emphasize that our primary objective was to establish a theoretical framework capable of elucidating the underlying causes of heavy-tailed spectral behavior (HT-MU). It is premature to precisely link model generalization to our hyperparameters, $\alpha$ and $\beta$ in (2) and $\kappa$ in (6). In future work, we can examine more closely this connection and extend the scaling law in Proposition 5.1 to more general cases.

## Acknowledgements

The authors thank Dr. Chris van der Heide for reading our manuscript and providing valuable feedback. LH is supported by the Australian Research Council through a Discovery Early Career Researcher Award (DE240100144). ZW is supported by the National Science Foundation under Grant No. DMS-1928930, as a resident at the Simons Laufer Mathematical Sciences Institute in Berkeley, California, in Spring 2025. MWM acknowledges partial support from DARPA, DOE, NSF, and ONR.

## Impact Statement

This paper presents work whose goal is to advance the field of Machine Learning. There are many potential societal consequences of our work, none of which we feel must be specifically highlighted here.

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

# Appendices

## Contents

## A. Experiment Details

### A.1. Empirical Spectral Distributions of NTK Matrices

In this section, we describe the experimental details behind the results in Figure 1, where we plot the empirical spectral distributions of NTK Gram matrices. The (empirical) NTK is

$$k_{\mathrm{NTK}}(x, y) = Df_\Theta(x)^\top Df_\Theta(y) \in \mathbb{R}^{m \times m},$$

where $Df_\Theta \in \mathbb{R}^{d \times m}$ is the Jacobian of a function $f_\Theta : \mathcal{X} \to \mathcal{Y} \subset \mathbb{R}^m$ with respect to its parameters $\Theta \in \mathbb{R}^d$. The corresponding Gram matrix over $\{x_i\}_{i=1}^n \subset \mathcal{X}$ is the matrix $J \in \mathbb{R}^{mn \times mn}$ with $m \times m$ blocks $J_{ij} = k_{\mathrm{NTK}}(x_i, x_j)$. This

Gram matrix is often enormous when $f_\Theta$ is a deep neural network: in double precision, the empirical NTK Gram matrix for an ImageNet-1K classifier requires more memory than most international datacenters (Ameli et al., 2025). While some approximations exist to estimate certain quantities of this matrix (Mohamadi et al., 2023), none of these approximations are valid for the purposes of plotting the eigenspectrum. Instead, we consider models trained on a subsample of 1000 datapoints from the CIFAR-10 dataset with 10 classes (Krizhevsky, 2009). We remark that it is important, for our purposes, to ensure that the model is only trained on data contained in the matrix, as this will inevitably impact the shape of the resulting spectrum in our analysis.

In Figure 1, we consider three convolutional neural networks of moderate size: VGG11 (9.2M parameters) (Simonyan & Zisserman, 2014); ResNet9 (4.8M parameters); and ResNet18 (11.1M parameters) (He et al., 2016). The output layer of each model is altered from its usual ImageNet counterparts to be classified into ten classes. All networks were initialized with weights randomly chosen according to the standard He initialization scheme (He et al., 2016). The top half of Figure 1 plots the eigenspectrum of these networks at initialization over 1000 entries of the CIFAR-10 dataset. This is a similar scenario to that of Wang et al. (2023), but with real model architectures. The inverse Gamma law is plainly visible here across the spectrum; a maximum likelihood fit to $\alpha, \beta$ in the law $p(\lambda) \propto \lambda^{-\alpha} e^{-\beta/\lambda}$ is performed, with the Kolmogorov-Smirnov statistics reported[5]. Values of $\beta$ are small, and the tail exponent $\alpha < 2$. We remark that the corresponding weight matrices in this scenario exhibit spectral densities with the Marchenko-Pastur law.

Next, the networks are trained on this set of 1000 examples using 200 epochs of a cosine annealing learning rate schedule with a 200 epoch period, starting from a learning rate of 0.05 with a batch size of 64. All models achieve near-zero loss under cross-entropy and an accuracy of 100%. The lower half of Figure 1 presents the spectral densities after training. Here, there are two distinct "bumps", both of which exhibit inverse Gamma behavior. The right tail is not too different from that of the spectrum at initialization, so we draw special attention to the inverse Gamma behavior near zero. Once again, we perform maximum likelihood fits, with the Kolmogorov-Smirnov statistics reporting good adherence to the inverse Gamma law.

### A.2. Empirical Spectral Distributions of Trained Weight Matrices

In this section, we present the experimental details of Figure 2 for 5+1 phases of trained weight matrices and their empirical spectral distributions. In these experiments, we train a MiniAlexNet (Martin & Mahoney, 2019) for the `CIFAR-10` classification task with different batch sizes. This MiniAlexNet is a simplified version of AlexNet (Krizhevsky et al., 2012), which comprises two convolutional blocks–Conv($3 \to 64$, $3 \times 3$, stride $= 2$)+ReLU+MaxPool and Conv($64 \to 192$, $3 \times 3$)+ReLU+MaxPool–followed by two fully-connected layers ($192 \times 4 \times 4 \to 1000 \to 10$) with ReLU activations. All weights are initialized via the parametrization (Jacot et al., 2018), i.e., the entries of trainable weights are i.i.d. $\mathcal{N}(0, 1/\sqrt{\text{fan}_{\text{in}}})$. We normalize each data point in `CIFAR-10` to zero mean and unit variance per channel, and rescale it so that each flattened input vector has Euclidean norm $\sqrt{d}$ where $d = 3 \times 32 \times 32$ is the input dimension for each data point. We train this MiniAlexNet with SGD (momentum 0.9, weight decay $10^{-4}$), using a learning rate scaled as $2/\text{batch\_size}$ for different batch sizes and early stopping once the training loss falls below 0.01 or 200 epochs. In Figure 2, we separately train MiniAlexNet with batch sizes $1000, 800, 250, 100, 50$ and $5$; and we repeat the experiments 3 times for the average. Post-training, Figure 2 shows histograms of the eigenvalues of $WW^\top \in \mathbb{R}^{1000 \times 1000}$ where $W$ is the trained weight matrix in the first fully-connected layer with input dimension $192 \times 4 \times 4$ and output dimension 1000, for different batch sizes. Notice that, because of the NTK parameterization at initialization, the empirical spectral distribution of $WW^\top$ is close to the $\mathbf{MP}_\gamma$ with $\gamma = 1000/(192 \times 4 \times 4)$. For large batch sizes, batch\_size $= 1000$, after training, the empirical spectral distribution of $WW^\top$ remains similar to a random case, as depicted in Figure 2(a). The red dashed line is computed using the density function of $\mathbf{MP}_\gamma$ with $\gamma = 1000/(192 \times 4 \times 4)$, which is described by (E.3). This $\mathbf{MP}_\gamma$ can also be interpreted as $\mathbf{HTMP}_{\gamma,\kappa}$ with $\kappa = \infty$. When the batch size decreases, we observe eigenvalue bleed out from the bulk spectrum and the emergence of many spike eigenvalues in Figure 2(b) and (c). Despite this, the bulk spectrum remains relatively close to $\mathbf{MP}_\gamma$. Further decreasing the batch size, Figure 2(d) and (e), the bulk spectrum of the trained weight is also changed into heavier tailed distribution which is close to $\mathbf{HTMP}_{\gamma,\kappa}$ for some finite $\kappa$. The red dashed lines in Figure 2(d) and (e) are computed by the density function of $\mathbf{HTMP}_{\gamma,\kappa}$ with $\kappa = 5.5, 1.9$ and $\gamma = 1000/(192 \times 4 \times 4)$, defined in (E.5). Additionally, when the batch size is small (batch\_size $= 5$), the trained weight matrix exhibits significant rank deficiency, corresponding to $\mathbf{HTMP}_{\gamma,\kappa}$ with $\kappa$ approaching zero. In such scenarios, the model's ability to effectively learn the task is compromised due to over-regularization. Hence, Figure 2(f) is not a particularly interesting case for further exploration.

---

[5]Recall that the Kolmogorov-Smirnov statistic between two cumulative distribution functions $F_1$ and $F_2$ is $d_{\text{KS}}(F_1, F_2) = \sup_{x \in \mathbb{R}} |F_1(x) - F_2(x)|$.

## B. Stochastic Minimization Model Optimizes PAC-Bayes Bounds

In this section, we outline the connection between our stochastic minimization operator, $\mathrm{smin}_\Theta$ of Equation (1), and the celebrated PAC-Bayesian bounds (McAllester, 1998), which have seen increasing popularity and success for explaining the success of deep learning (Wilson, 2025) and for providing non-vacuous generalization bounds (Lotfi et al., 2022). In particular, we will show that the stochastic minimization operator applied to *training losses* (as Monte–Carlo estimators of the *test loss*) minimizes, by definition, a PAC-Bayesian upper bound on the test loss. Our discussion mostly follows expositions by Germain et al. (2016). For the sake of brevity, instead of the two-part loss function, $L(\Theta, \Phi)$ from Section 3, we consider arbitrary loss functions of the form $L(\Theta)$. This is no less general: we can once again split the parameter space into two parts and apply our arguments here to each part independently.

The foundation of modern PAC-Bayes proofs, as first outlined in Bégin et al. (2016), is the Donsker–Varadhan variational formula, first introduced in Donsker & Varadhan (1983, Eqn. (1.7)). For completeness, we state both the formula and its proof in Lemma B.1.

**Lemma B.1** (DONSKER–VARADHAN VARIATIONAL FORMULA). *For any probability measure $\mu$ on a measurable space $\Xi$ and $f : \Xi \to \mathbb{R}$,*

$$\sup_{\nu \ll \mu} (\mathbb{E}_{\nu(\Theta)} f(\Theta) - d_{\mathrm{KL}}(\nu \| \mu)) = \log \mathbb{E}_{\mu(\Theta)} e^{f(\Theta)},$$

*and for $e^f \in L^1(\mu)$, the probability measure $\nu$ that achieves this equality satisfies*

$$\frac{\mathrm{d}\nu^*}{\mathrm{d}\mu}(\Theta) \propto \exp(f(\Theta)), \qquad \Theta \in \Xi. \tag{B.1}$$

*Proof.* For brevity, let $\mu = \pi_\Theta$. Let $\nu$ denote an arbitrary probability measure that is absolutely continuous with respect to $\mu$. By the Lebesgue decomposition, $\mu = \mu_{\mathrm{ac}} + \mu_{\mathrm{sing}}$ where $\mu_{\mathrm{ac}} \ll \nu$ and $\mu_{\mathrm{sing}} \perp \nu$. Then, the Radon–Nikodym derivative $\frac{\mathrm{d}\nu}{\mathrm{d}\mu} = \frac{\mathrm{d}\nu}{\mathrm{d}\mu_{\mathrm{ac}}}$ $\nu$-almost everywhere, and

$$d_{\mathrm{KL}}(\nu \| \mu) = \mathbb{E}_{\nu(\Theta)} \log \frac{\nu(\Theta)}{\mu(\Theta)} = \mathbb{E}_{\nu(\Theta)} \log \frac{\nu(\Theta)}{\mu_{\mathrm{ac}}(\Theta)} = d_{\mathrm{KL}}(\nu \| \mu_{\mathrm{ac}}).$$

Furthermore, by Jensen's inequality,

$$\mathbb{E}_{\nu(\Theta)} \left( e^{f(\Theta)} \frac{\mathrm{d}\mu_{\mathrm{ac}}}{\mathrm{d}\nu} \right) \le \log \mathbb{E}_{\nu(\Theta)} e^{f(\Theta)} \frac{\mathrm{d}\mu_{\mathrm{ac}}}{\mathrm{d}\nu}(\Theta) = \mathbb{E}_{\mu_{\mathrm{ac}}(\Theta)} \exp(f(\Theta)) \le \mathbb{E}_{\mu(\Theta)} \exp(f(\Theta)).$$

At the same time,

$$\mathbb{E}_{\nu(\Theta)} \left( e^{f(\Theta)} \frac{\mathrm{d}\mu_{\mathrm{ac}}}{\mathrm{d}\nu} \right) = \mathbb{E}_{\nu(\Theta)} f(\Theta) - d_{\mathrm{KL}}(\nu \| \mu_{\mathrm{ac}}) = \mathbb{E}_{\nu(\Theta)} f(\Theta) - d_{\mathrm{KL}}(\nu \| \mu),$$

and so

$$\sup_{\nu \ll \mu} (\mathbb{E}_{\nu(\Theta)} f(\Theta) - d_{\mathrm{KL}}(\nu \| \mu)) \le \mathbb{E}_{\mu(\Theta)} \exp(f(\Theta)).$$

It now only remains to show that the measure $\nu^*$ satisfying (B.1) saturates this inequality. Letting $\mathcal{Z} = \mathbb{E}_{\mu(\Theta)} \exp(f(\Theta))$, so that $\frac{\mathrm{d}\nu^*}{\mathrm{d}\mu} = \mathcal{Z}^{-1} \exp(f(\Theta))$, there is

$$d_{\mathrm{KL}}(\nu^* \| \mu) = \mathbb{E}_{\nu^*(\Theta)} \log \left( \frac{\mathrm{d}\nu^*}{\mathrm{d}\mu} \right) \mathrm{d}\nu^* = \mathbb{E}_{\nu^*(\Theta)} (f(\Theta) - \log \mathcal{Z}) = \mathbb{E}_{\nu^*(\Theta)} f(\Theta) - \log \mathcal{Z}.$$

Hence,

$$\mathbb{E}_{\nu^*(\Theta)} f(\Theta) - d_{\mathrm{KL}}(\nu^* \| \mu) = \log \mathcal{Z} = \log \mathbb{E}_{\mu(\Theta)} e^{f(\Theta)},$$

as required. $\qquad\square$

Now we return to the case where $\nu$ and $\mu$ are both absolutely continuous measures with respect to Lebesgue measure, and are in turn represented by probability densities on $\Xi \subset \mathbb{R}^d$, and we let $\mu = \pi_\Theta(\Theta)$, in line with the setup from Section 3. Most losses to be minimized are *training losses* of the form $L_n(\Theta) = n^{-1} \sum_{i=1}^n \ell(\Theta, x_i, y_i)$, where each $(x_i, y_i)$ is an

input-output pair. Training losses are *random*; a Monte-Carlo approximation of the test loss $L(\Theta)$ defined as the expectation $\mathbb{E}L_n(\Theta)$. Let $\tau > 0$ and consider the function $f_n(\Theta) = \frac{n}{\tau}(L(\Theta) - L_n(\Theta))$. Then, by Lemma B.1,

$$\mathbb{E}_{q(\Theta)} f_n(\Theta) \leq d_{\mathrm{KL}}(q\|\pi_\Theta) + \log \zeta_{n,\tau},$$

where $\zeta_{n,\tau}$ is defined by

$$\zeta_{n,\tau} = \mathbb{E}_{\pi_\Theta(\Theta)} f_n(\Theta) = \mathbb{E}_{\pi_\Theta(\Theta)} \exp\left(\frac{n}{\tau}(L(\Theta) - L_n(\Theta))\right).$$

If $(x_i, y_i)_{i=1}^n$ are independent and identically distributed, then subgaussian concentration inequalities such as the Bernstein inequality (Boucheron et al., 2013, Theorem 2.10) imply that $\mathbb{E}\zeta_{n,\tau}$ is bounded as $n \to \infty$. In particular, we may assume that

$$K_\tau := \sup_n \mathbb{E}\zeta_{n,\tau} < +\infty.$$

From Markov's inequality, for $0 < \delta < 1$, $\mathbb{P}(\zeta_{n,\tau} > K_\tau/\delta) \leq \mathbb{P}(\zeta_{n,\tau} > \mathbb{E}\zeta_{n,\tau}/\delta) \leq \delta$, and so with probability at least $1 - \delta$,

$$\mathbb{E}_{q(\Theta)} f_n(\Theta) \leq d_{\mathrm{KL}}(q\|\pi_\Theta) + \log\left(\frac{K_\tau}{\delta}\right).$$

Recalling the definition of $f_n$ reveals the (simplified) *PAC-Bayes bound*, as presented in Germain et al. (2009, Theorem 2.1).

**Theorem B.2** (PAC-BAYES THEOREM (Germain et al., 2009)). *For any probability density $q$ absolutely continuous with respect to $\pi_\Theta$ on $\Xi \subset \mathbb{R}^d$, assuming $(x_i, y_i)_{i=1}^n$ are iid, then with probability at least $1 - \delta$,*

$$\mathbb{E}_{q(\Theta)} L(\Theta) \leq \mathbb{E}_{q(\Theta)} L_n(\Theta) + \frac{\tau}{n} d_{\mathrm{KL}}(q\|\pi_\Theta) + \frac{\tau}{n} \log\left(\frac{K_\tau}{\delta}\right). \tag{B.2}$$

As the left-hand-side of (B.2) corresponds with the test loss, we are interested in considering densities $q$ that assign probabilities to model parameters $\Theta$ that have been fitted to the data. The density $\pi_\Theta$ acts as the *prior* in the PAC-Bayes theory, and it assigns probability mass to models that are believed to be more or less likely to exhibit small test loss, prior to looking at data. This is to be contrasted with uniform PAC bounds, which typically treat all models as equally likely. A key feature of the prior is that it is data-independent, but otherwise it is arbitrary. It is this feature that has allowed (B.2) to achieve the tightest generalization bounds to date (Lotfi et al., 2022).

It is now also immediately clear, by definition, that the density $q_n$ which minimizes the bound of the test error on the right-hand side of (B.2) is precisely given by the argsmin operator:

$$q_n(\Theta) = \underset{\Theta}{\overset{\pi_\Theta, \tau_n}{\mathrm{argsmin}}}\, L_n(\Theta), \qquad \text{where } \tau_n := \frac{\tau}{n}.$$

## C. Proofs of Assorted Results from the Main Text

In this section, we present proofs of several results from the main text. Below, we first prove Proposition 3.1 and subsequently complete the proof of Equation (3) for activation matrices introduced in Section 3.2. We defer the analysis of NTK matrices and the proof of Equation (5) to Appendix D for further details. Subsequently, we present the proofs of Propositions 5.1, 5.3, and 5.3. Our random matrix results, as outlined in Theorem 4.1 and Theorem 4.3, are presented in Section E to ensure clarity and self-containedness.

### C.1. Proof of Proposition 3.1

The primary tool in the proof of Proposition 3.1 is a corollary of the Donsker–Varadhan variational formula in Lemma B.1, which we now state in the notation of Section 3.

**Corollary C.1.** *For $\Xi \subseteq \mathbb{R}^d$ and $f : \Xi \to \mathbb{R}$ such that $q(\Theta) = \exp(-\frac{1}{\tau}f(\Theta))\pi_\Theta(\Theta)$ is integrable over $\Theta \in \Xi$, $\mathrm{argsmin}_\Theta^{\pi_\Theta, \tau} f(\Theta) \propto q(\Theta)$.*

*Proof.* Letting $\mathcal{P}$ denote the set of probability densities that are absolutely continuous with respect to Lebesgue measure

on $\Xi$,

$$\underset{\Theta}{\overset{\pi_{\Theta,\tau}}{\text{argsmin}}} f(\Theta) = \underset{q(\Theta)\in\mathcal{P}}{\text{argmin}}[\mathbb{E}_{q(\Theta)}f(\Theta) + \tau d_{\mathrm{KL}}(q\|\pi_\Theta)]$$

$$= \underset{q(\Theta)\in\mathcal{P}}{\text{argmax}}\left[\mathbb{E}_{q(\Theta)}\left(-\frac{1}{\tau}f(\Theta)\right) - d_{\mathrm{KL}}(q\|\pi_\Theta)\right].$$

Applying Lemma B.1, it follows that the minimizing density is proportional to $q(\Theta)$. $\qquad\square$

Let $q$ be as in Corollary C.1, and let $\mathcal{Z}_\tau = \mathbb{E}_{\Theta\sim\pi_\Theta}\exp(-\frac{1}{\tau}f(\Theta))$, so that $\tilde{q}(\Theta) = \mathcal{Z}_\tau^{-1}q(\Theta)$ is a probability density. As an immediate consequence of Lemma B.1, since $d_{\mathrm{KL}}(\tilde{q}\|\pi_\Theta) = -\frac{1}{\tau}\mathbb{E}_{\Theta\sim\tilde{q}}f(\Theta) - \log\mathcal{Z}_\tau$, it follows that

$$\underset{\Theta}{\overset{\pi_{\Theta,\tau}}{\text{smin}}} f(\Theta) = -\tau\log\mathcal{Z}_\tau. \tag{C.1}$$

Proposition 3.1 now follows immediately from Corollary C.1 and (C.1): for $\mathcal{Z}_\tau(\Phi)$ as defined in Proposition 3.1,

$$\underset{\Phi}{\overset{\pi_{\Phi,\eta}}{\text{argsmin}}}\,\underset{\Theta}{\overset{\pi_{\Theta,\tau}}{\text{smin}}}\, L(\Theta,\Phi) \propto \exp\left(\frac{\tau}{\eta}\log\mathcal{Z}_\tau(\Phi)\right)\pi_\Phi(\Phi) = \mathcal{Z}_\tau(\Phi)^{\tau/\eta}\pi_\Phi(\Phi).$$

## C.2. Proof of Equation (3): for Activation Matrices

Now we prove Equation (3). Recall that

$$L(W,\Phi) = \|\Phi W - Y\|_F^2 + \mu\|W\|_F^2$$

where $\mu > 0$. Our objective is to first compute

$$\mathcal{Z}_\tau(\Phi) = \mathbb{E}_{W\sim\mathcal{N}(0,\sigma^2 I)}\exp\left(-\frac{1}{\tau}\|\Phi W - Y\|_F^2 - \frac{\mu}{\tau}\|W\|_F^2\right).$$

For the sake of brevity, we ignore all constants of proportionality that do not depend on $\Phi$. First, observe that

$$\mathcal{Z}_\tau(\Phi) \propto \int\exp\left(-\frac{1}{\tau}\|\Phi W - Y\|_F^2 - \frac{\mu}{\tau}\|W\|_F^2 - \frac{1}{2\sigma^2}\|W\|_F^2\right)\mathrm{d}W,$$

and so $\mathcal{Z}_\tau(\Phi)$ is proportional to the marginal likelihood $p(Y|\Phi) = \int p(Y|W,\Phi)p(W)\mathrm{d}W$ corresponding to the likelihood

$$p(Y|W,\Phi) = \frac{1}{(2\pi\tau)^{mn/2}}\exp\left(-\frac{1}{\tau}\|\Phi W - Y\|_F^2\right),$$

of the model $Y = \Phi W + \frac{\tau}{2}\epsilon$, and the prior

$$p(W) \propto \exp\left(-\left(\frac{\mu}{\tau} + \frac{1}{2\sigma^2}\right)\|W\|_F^2\right),$$

implying $W\sim\mathcal{N}(0,\tilde{\sigma}^2 I)$, where

$$\tilde{\sigma}^2 = \left(\frac{2\mu}{\tau} + \frac{1}{\sigma^2}\right)^{-1} = \frac{\sigma^2}{1 + \frac{2\mu\sigma^2}{\tau}}.$$

Let $y = \text{vec}(Y)$ and $w = \text{vec}(W)$, recalling that $y = (I\otimes\Phi)w + \frac{\tau}{2}\epsilon$ under the model likelihood. Since $w$ is normal, $y|\Phi$ is normal also, with $\mathbb{E}[y|\Phi] = 0$ and

$$\begin{aligned}\text{Cov}(y) &= \text{Cov}((I\otimes\Phi)w) + \frac{\tau}{2}I \\ &= (I\otimes\Phi)(\tilde{\sigma}^2 I)(I\otimes\Phi^\top) + \frac{\tau}{2}I \\ &= I\otimes(\tilde{\sigma}^2\Phi\Phi^\top + \frac{\tau}{2}I).\end{aligned}$$

Therefore,

$$p(y|\Phi) = \frac{1}{(2\pi)^{nm/2}} \cdot \frac{\exp\left(-\frac{1}{2} y^\top [I \otimes (\tilde{\sigma}^2 \Phi\Phi^\top + \frac{\tau}{2} I)]^{-1} y\right)}{\det(I \otimes (\tilde{\sigma}^2 \Phi\Phi^\top + \frac{\tau}{2} I))^{1/2}}.$$

Since $\mathrm{tr}(A^\top B) = \mathrm{vec}(A)^\top \mathrm{vec}(B)$, we can get

$$y^\top [I \otimes (\tilde{\sigma}^2 \Phi\Phi^\top + \tfrac{\tau}{2} I)]^{-1} y = y^\top \mathrm{vec}((\tilde{\sigma}^2 \Phi\Phi^\top + \tfrac{\tau}{2} I)^{-1} Y)$$
$$= \mathrm{tr}(Y^\top (\tilde{\sigma}^2 \Phi\Phi^\top + \tfrac{\tau}{2} I)^{-1} Y).$$

In terms of $Y$,

$$\mathcal{Z}_\tau(\Phi) \propto p(Y|\Phi) = \frac{\exp\left(-\frac{1}{2}\mathrm{tr}(Y^\top (\tilde{\sigma}^2 \Phi\Phi^\top + \frac{\tau}{2} I)^{-1} Y)\right)}{\det(\tilde{\sigma}^2 \Phi\Phi^\top + \frac{\tau}{2} I)^{m/2}}.$$

### C.3. Proof of Proposition 5.1

Recall the solution to the ridge regression problem as

$$\hat{w} := (\Phi^\top \Phi + \lambda I)^{-1} \Phi^\top Y \in \mathbb{R}^d, \tag{C.2}$$

and, by definition, we know that

$$\hat{w} = (\Phi^\top \Phi + \lambda I)^{-1} \Phi^\top \Phi w_*,$$

for any $\lambda > 0$. Thus, the training loss is

$$L_0 := \frac{1}{n} \sum_{i=1}^n (\varphi(x_i)^\top \hat{w} - \varphi(x_i)^\top w_*)^2] = \lambda^2 w_*^\top (\Phi^\top \Phi + \lambda I)^{-1} \Phi^\top \Phi (\Phi^\top \Phi + \lambda I)^{-1} w_*. \tag{C.3}$$

Here, we do not consider label noise: each label is the form of $y_i = \varphi(x_i)^\top w_*$. Based on the definition of the generalization error defined in Proposition 5.1, we have

$$L = \mathbb{E}[(\varphi(x)^\top \hat{w} - \varphi(x)^\top w_*)^2] \tag{C.4}$$
$$= (\hat{w} - w_*)^\top \Sigma (\hat{w} - w_*) \tag{C.5}$$
$$= w_*^\top ((\Phi^\top \Phi + \lambda I)^{-1} \Phi^\top \Phi - I) \Sigma ((\Phi^\top \Phi + \lambda I)^{-1} \Phi^\top \Phi - I) w_* \tag{C.6}$$
$$= \lambda^2 w_*^\top (\Phi^\top \Phi + \lambda I)^{-1} \Sigma (\Phi^\top \Phi + \lambda I)^{-1} w_*, \tag{C.7}$$

where we denote that $\Sigma := \mathbb{E}[\varphi(x)\varphi(x)^\top]$. Suppose that $w_* \sim \mathcal{N}(0, \frac{1}{d} I)$. Then, we can apply Hanson-Wright inequality to get

$$L = \frac{\lambda^2}{d} \mathrm{tr}(\Phi^\top \Phi + \lambda I)^{-2} \Sigma + o(1),$$

with high probability as $d \to \infty$. Based on our assumptions in Proposition 5.1, we set $\Sigma$ to be the identity matrix, namely the isotropic case. (For general $\Sigma$, we would need to apply Theorem 4.3 and a deterministic equivalent statement for $\mathrm{tr}(\Phi^\top \Phi + \lambda I)^{-2} A$, for any deterministic matrix $A \in \mathbb{R}^{d \times d}$. However, currently, such a deterministic equivalence for inverse Wishart matrices is unknown in RMT. We leave this general statement for future work.)

Since we assume that $\Phi^\top \Phi$ has a limiting eigenvalue distribution with density functions described in Theorem 4.1, we can claim that

$$L \to \lambda^2 S'_{\rho_{\gamma,\kappa}}(-\lambda) =: f(\lambda),$$

almost surely, as $d \to \infty$, where $S'_{\rho_{\gamma,\kappa}}(z)$ is the derivative of $S_{\rho_{\gamma,\kappa}}(z)$, defined in Proposition E.3, with respect to $z$. Here $S_{\rho_{\gamma,\kappa}}(z)$ is the Stieltjes transform of $\mathbf{HTMP}_{\gamma,\kappa}$ introduced in Theorem 4.1. For more details, see Appendix E. From (E.7), we know that

$$S_{\rho_{\gamma,\kappa}}(-\lambda) = \frac{\kappa}{2\gamma} \cdot \frac{U(\kappa/2 + 1, 2 - \frac{\kappa}{2\gamma} + \frac{\kappa}{2}; \frac{\kappa\lambda}{2\gamma})}{U(\kappa/2, -\frac{\kappa}{2\gamma} + 1 + \frac{\kappa}{2}; \frac{\kappa\lambda}{2\gamma})}, \quad \lambda > 0. \tag{C.8}$$

Now we want to study the asymptotic limit of the function $f(\lambda)$ as $\lambda \to 0$. Notice that $\lambda \to 0$ represents the ridgeless regression; and, in this case, the training loss will converge to zero, because of (C.3). Hence, we are considering the case when the estimator interpolates all the training datasets.

From DLMF, (13.3.22), recall that

$$\frac{\mathrm{d}}{\mathrm{d}z}U(a,b,z) = -aU(a+1,b+1,z).$$

In general, from DLMF, (13.2.16-22), we also know that

$$U(a,b,z) = \frac{\Gamma(b-1)}{\Gamma(a)}z^{1-b} + O\left(z^{2-\Re b}\right), \quad \Re b \geq 2, \, b \neq 2, \tag{C.9}$$

$$U(a,2,z) = \frac{1}{\Gamma(a)}z^{-1} + O(\ln z), \tag{C.10}$$

$$U(a,b,z) = \frac{\Gamma(b-1)}{\Gamma(a)}z^{1-b} + \frac{\Gamma(1-b)}{\Gamma(a-b+1)} + O\left(z^{2-\Re b}\right), \quad 1 \leq \Re b < 2, \, b \neq 1, \tag{C.11}$$

$$U(a,1,z) = -\frac{1}{\Gamma(a)}(\ln z + \psi(a) + 2\gamma) + O(z \ln z), \tag{C.12}$$

$$U(a,b,z) = \frac{\Gamma(1-b)}{\Gamma(a-b+1)} + O\left(z^{1-\Re b}\right), \quad 0 < \Re b < 1, \tag{C.13}$$

$$U(a,0,z) = \frac{1}{\Gamma(a+1)} + O(z \ln z), \tag{C.14}$$

$$U(a,b,z) = \frac{\Gamma(1-b)}{\Gamma(a-b+1)} + O(z), \quad \Re b \leq 0, \, b \neq 0. \tag{C.15}$$

Now for simplicity, in (C.8), we define that

$$z = \frac{\kappa\lambda}{2\gamma},$$

$$a = \kappa/2 + 1, \quad b = \kappa/2 + 2 - \frac{\kappa}{2\gamma},$$

$$c = \kappa/2, \quad d = -\frac{\kappa}{2\gamma} + 1 + \frac{\kappa}{2}.$$

Thus, we now have

$$\begin{aligned}
f(\lambda) &= z^2 \frac{cU(c+1,d+1;z)U(a,b;z) - aU(a+1,b+1;z)U(c,d;z)}{U(c,d;z)^2} \\
&= z^2 \frac{cU(a,b;z)^2 - aU(a+1,b+1;z)U(c,d;z)}{U(c,d;z)^2}.
\end{aligned}$$

Let us consider the case when $b > 1$ and $d > 0$. Hence, using the asymptotic results in (C.9)-(C.15), we can get that

$$f(\lambda) \asymp z^{d+1},$$

as $z \to 0+$. Notice that here $a, b, c, d$ are constants and $\gamma \in (0,1)$. Therefore, when assuming that $\lambda = n^{-\ell}$, we can conclude the result in Proposition 5.1. For more comparison of Proposition 5.1 with previous literature, we refer to Appendix H.

## C.4. Proof of Proposition 5.2

The stochastic gradients $\hat{\nabla}L_N$ are given by $\hat{\nabla}L_N = Df_{\Theta,\Phi}(X)^\top Z$, where $Z = f_{\Theta,\Phi}(X) - Y$ are assumed to be independent and standard normal. Consequently,

$$\|\hat{\nabla}L_N\|^2 = Z^\top Df_{\Theta,\Phi}(X)Df_{\Theta,\Phi}(X)^\top Z = Z^\top JZ,$$

where $J$ has the inverse-Wishart distribution. Due to a result of Hotelling (1992), $\frac{d-N+1}{dN}\|\hat{\nabla}L_N\|^2$ is $F$-distributed with $(N, d-N+1)$ degrees of freedom, and therefore satisfies (10).

### C.5. Proof of Proposition 5.3

Consider the activation matrix model in Section 3. By Golub & Pereyra (1973, Theorem 1), with the pseudo inverse, trained weights are given by $W = \Phi^\dagger Y$, so $A = d^{-1} W^\top W = d^{-1} Y^\top (\Phi \Phi^\top)^{-1} Y$. Since the (nonzero) eigenvalues of the product of two matrices $AB$ are always the same as those of $BA$, assuming $\Sigma = YY^\top$ is full rank, the spectrum of $A^{-1}$ is equivalent to that of $B = \Sigma^{-1} M$ where $M = \Phi \Phi^\top$. Under the conditions of Theorem 4.1, $M$ satisfies the Master Model (2) and so

$$q(M) \propto (\det M)^{-\alpha} \exp(-\beta \mathrm{tr}(\Sigma M^{-1})) \pi(M).$$

Performing the change of variables $M \mapsto \Sigma^{-1} M$, we find that

$$q(B) \propto (\det B)^{-\alpha} \exp(-\beta \mathrm{tr}(B^{-1})) \pi(\Sigma B),$$

and since we assert that $\pi(\Sigma B)$ has a Weyl formula that aligns with (6) for $V \equiv 1$, the eigenvalues of $B$ satisfy

$$q(\lambda_1, \dots, \lambda_N) \propto \prod_{i=1}^{N} \lambda_i^{-\alpha} e^{-\beta/\lambda_i} \prod_{\substack{i,j=1,\dots,N \\ i<j}} |\lambda_i - \lambda_j|^{\kappa/N}.$$

From Theorem 4.1, with $B = B(m,d)$ and $\gamma = \frac{\kappa/2}{\alpha - \kappa/2 - 1}$, the sequence of empirical spectral distributions satisfy

$$\mathrm{spec}\left( \frac{\beta}{\alpha - \kappa/2 - 1} B(m,d)^{-1} \right) \xrightarrow{m,d\to\infty} \mathbf{HTMP}_{\gamma,\kappa}.$$

Since $\mathrm{spec}(B(m,d)^{-1}) = \mathrm{spec}(A(m,d))$, the result follows.

## D. NTK Matrix Densities using the Interpolating Information Criterion

Motivated by the overparameterization ($d > n$) of many deep neural network models, Hodgkinson et al. (2023) developed approximations to the marginal likelihood, for very general models, that are effective in the interpolating regime. These approximations led to the development of an *interpolating information criterion* to judge model quality in the interpolating regime. In this section, we will show how their techniques combined with Proposition 3.1 reveal a precise family of densities for the NTK matrices of interpolating models. We will also show how the NTK approximation discussed in Section 3 compares and its implications on implicit regularization.

Suppose that $\pi_\Theta(\Theta) \propto e^{-\frac{1}{\mu} R_\Phi(\Theta)}$, and let $\Theta_0 = \mathrm{argmin}_\Theta R_\Phi(\Theta)$ and $\bar{R}_\Phi(\Theta) = R_\Phi(\Theta) - R_\Phi(\Theta_0)$. Consider the *neural tangent kernel matrix*, $J(\Phi) \in \mathbb{R}^{mn \times mn}$, which satisfies $J(\Phi)_{ij} = Df(x_i)^\top Df(x_j)$, where $Df \in \mathbb{R}^{d \times m}$ is the Jacobian of a model, $f_{\Theta,\Phi} : \mathcal{X} \to \mathcal{Y} \subset \mathbb{R}^m$, in its parameters $\Theta \in \mathbb{R}^d$. If $\nabla^2_{\mathcal{M}}$ represents the manifold Hessian (Absil et al., 2008), then the marginal Gibbs likelihood for the loss

$$L(\Theta, \Phi) = \|f_{\Theta,\Phi}(X) - Y\|_F^2, \tag{D.1}$$

for inputs $X$ and outputs $Y$, satisfies (Hodgkinson et al., 2023, Theorem 1)

$$\mathcal{Z}_\tau(\Phi) \propto \frac{e^{-\frac{1}{\mu} \bar{R}_\Phi(\Theta^*)}}{\sqrt{\det J(\Phi)}} \sqrt{\frac{\det \nabla^2 R_\Phi(\Theta_0)}{\det \nabla^2_{\mathcal{M}} R_\Phi(\Theta^*)}} [1 + \mathcal{O}(\tau + \mu)], \tag{D.2}$$

where $\Theta^*$ is the interpolator

$$\Theta^* = \mathrm{argmin}_\Theta R_\Phi(\Theta) \quad \text{subject to} \quad L(\Theta, \Phi) = 0. \tag{D.3}$$

Using (D.2) with Proposition 3.1 constitutes a precise formula for minimizing the density: up to an $\mathcal{O}(\tau + \mu)$ factor,

$$q(\Phi) \propto (\det J(\Phi))^{-\rho/2} \left( \frac{\det \nabla^2 R_\Phi(\Theta_0)}{\det \nabla^2_{\mathcal{M}} R_\Phi(\Theta^*)} \right)^{\rho/2} \exp\left( -\frac{\rho}{\mu} \bar{R}_\Phi(\Theta^*) \right) \pi_\Phi(\Phi). \tag{D.4}$$

While we cannot elucidate the spectral density from (D.4), it provides a more accurate picture of the complexities with ascertaining the densities of trained feature matrices at scale. Indeed, we can use (D.4) to understand our model for the NTK feature matrix (5) without appealing to linear approximations.

First, we ask the question: if $\Theta^*$ should yield a robust predictor, what is a good choice of $R_\Phi$? For the sake of brevity, we drop the dependence on the features $\Phi$ and write $R_\Phi$ as simply $R$. In the overparameterized regime, the test loss is primarily dictated by the variance in model predictions with respect to label noise (Holzmüller, 2020). For a Lipschitz loss function, this variance depends on that of the estimator $\Theta^*$ itself. Consequently, to ensure that a solution to (D.3) exhibits small test loss, the regularizer $R$ should be chosen to control the variance of the induced estimator. This leads us to the following Definition D.1. Here, the variance of a random vector is considered as the sum of the variances over each coordinate, i.e., $\mathrm{Var}\,X = \sum_i \mathrm{Var}\,X_i$.

**Definition D.1** (IDEAL REGULARIZER)**.** An ideal regularizer (for minimizing test error) is a choice of $R(\theta)$ that controls the induced label noise of the corresponding estimator:

$$R(\Theta) \geq \frac{1}{2}\mathrm{Var}_Y\,\Theta^*(Y).$$

Although the precise form of an ideal regularizer may be quite complicated, a simple lower bound can be derived in the scenario where the label noise has finite covariance.

**Lemma D.2.** *If $J$ is full rank, then* $\mathrm{Var}_Y\,\Theta^*(Y) \geq \mathrm{tr}(\Sigma J^{-1}) + o(\|\Sigma\|_2)$, *where* $\Sigma = \mathrm{Cov}(Y)$.

*Proof.* First, performing a Taylor series expansion in $Y$, we find that

$$\mathrm{Var}_Y\,\Theta^*(Y) = \|D_Y\Theta^*(Y)\,\Sigma^{1/2}\|_F^2 + o(\|\Sigma\|_2), \tag{D.5}$$

where $\Sigma^{1/2}$ is the symmetric square root of $\Sigma$. Letting $f_{\Theta^*}$ denote the underlying model, since $f_{\Theta^*}(X) = Y$ in the interpolating regime, taking derivatives of both sides in $Y$ reveals that $DF\,D_Y\Theta^* = I$, and $D_Y\Theta^*$ is a right-inverse of $DF$. Consequently, $D_Y\Theta^*\Sigma^{1/2}$ is also a right-inverse of $\Sigma^{-1/2}DF$, and so

$$\begin{aligned}
\|D_Y\Theta^*\Sigma^{1/2}\|_F^2 &\geq \|(\Sigma^{-1/2}DF)^+\|_F^2 \\
&= \mathrm{tr}\left((\Sigma^{-1/2}DFDF^\top\Sigma^{-1/2})^{-1}\right) \\
&= \mathrm{tr}(\Sigma J^{-1}),
\end{aligned}$$

where the second equality follows from the identity $\|X^\dagger\|_F^2 = \mathrm{tr}((XX^\top)^{-1})$. Altogether, this implies that

$$\mathrm{Var}_Y\,\Theta^*(Y) \geq \mathrm{tr}(\Sigma J^{-1}) + o(\|\Sigma\|_2),$$

as required. □

*Remark* D.3. In the case where $Y$ is normally distributed, (Cacoullos, 1982, Proposition 3.7) implies that (D.5) can be replaced by

$$\mathrm{Var}_Y\,\Theta^*(Y) \geq \|D_Y\Theta^*\Sigma^{1/2}\|_F^2,$$

and so the limit $o(\|\Sigma\|_2)$ can be dropped.

Choosing $R(\Theta) = \mathrm{tr}(\Sigma J^{-1}) = \|DF(\Theta)^+Y\|_F^2$ to saturate these lower bounds, (D.4) becomes

$$q(\Phi) \propto (\det J(\Phi))^{-\rho/2}\left(\frac{\det \nabla^2 R(\Theta_0)}{\det \nabla^2_{\mathcal{M}} R(\Theta^*)}\right)^{\rho/2}\exp\left(-\frac{\rho}{\mu}\mathrm{tr}(\Sigma J(\Phi)^{-1})\right)\pi_\Phi(\Phi). \tag{D.6}$$

This is equivalent to the density (5) obtained under the linearized model except for the second-order factor $(\det \nabla^2 R(\Theta_0)/\det \nabla^2_{\mathcal{M}} R(\Theta^*))^{\rho/2}$. Formally speaking, it is unclear what the influence of this factor is for general models, however, we can make some conjectures. For (D.6) to coincide with (5) under the change of variables $\Phi \mapsto J(\Phi)$, it is important for $\nabla^2 R(\Theta_0)$ and $\det \nabla^2_{\mathcal{M}} R(\Theta^*)$ to be minimally dependent on $J(\Phi)$. In the former case, we could that the mutual information between $R(\Theta_0)$ and $R(\Theta^*)$ is small as a consequence of the training procedure. This has some empirical merit according to Fort et al. (2020). The latter object is more complex, but depends primarily on the null space of $DF(\Theta^*)$ and the higher-order derivatives of $F$. Again, we may conjecture that the mutual information between these quantities and the eigenspectrum of $J(\Phi)$ is relatively minimal, although such claims require further study.

# E. The High-Temperature Marchenko-Pastur (HTMP) Distribution

In this section, we present more background on the beta ensemble in RMT, we then prove Theorems 4.1 and 4.3, and we then present additional properties of the HTMP distribution introduced in Theorem 4.1.

## E.1. Beta Laguerre Ensembles for Two Regimes

In RMT, for $N \geq 1$ and $\beta > 0$, the beta-Hermite (Gaussian) ensemble of size $N \times N$ is defined by its joint eigenvalue density function:

$$p_{\mathrm{H}}(\lambda_1, \ldots, \lambda_N) \propto e^{-\frac{\beta}{2} \sum_{i=1}^N \lambda_i^2} \prod_{1 \leq i < j \leq N} |\lambda_j - \lambda_i|^\beta.$$

This reduces to the classical random matrix ensembles: GOE, GUE, and GSE, when $\beta = 1, 2, 4$ respectively. We refer to Chapter 4 of Anderson et al. (2009) for more details on beta ensemble matrices. In our setting, replacing the quadratic confinement above with a Wishart-type matrix yields the beta-Laguerre ensemble (Dumitriu & Edelman, 2002).

We define a random matrix $A_N$ with a beta-Laguerre ensemble of size $N \times N$ via its joint eigenvalue density function:

$$q(\lambda_1, \ldots, \lambda_N) \propto \prod_{i=1}^N \left( \lambda_i^{\frac{\beta}{2}(D-N+1)-1} e^{-\frac{\beta D}{2} \lambda_i} \right) \prod_{\substack{i,j=1,\ldots,N \\ i<j}} |\lambda_j - \lambda_i|^\beta, \tag{E.1}$$

for some $\beta = \beta(N) > 0$ and any $N, D \in \mathbb{N}$. Denote $\beta = \kappa/N$ for some $\kappa > 0$. In this section, we present RMT for the beta-Laguerre ensemble for two different regimes, based on different limits of $\beta$, or equivalently, the limits of $\kappa$. We clarify that the parameter $\beta$ in the beta-Laguerre ensemble, as defined in (E.1), is distinct from the $\beta$ encountered in our Master Model Ansatz, presented in (3.2). With slight ambiguity, the term $\beta$ is employed in this section for a distinct purpose compared to its usage in the main text.

We let $D, N \to \infty$ with $D \geq N$ and $N/D \to \gamma \in (0, 1)$. Then, (E.1) is asymptotically equivalent to

$$q(\lambda_1, \ldots, \lambda_N) \propto \prod_{i=1}^N \left( \lambda_i^{\frac{\kappa}{2}(\frac{1}{\gamma}-1)-1} e^{-\frac{\kappa}{2\gamma} \lambda_i} \right) \prod_{\substack{i,j=1,\ldots,N \\ i<j}} |\lambda_j - \lambda_i|^{\kappa/N}. \tag{E.2}$$

We will prove Theorem 4.1 based on the joint density function (E.2). Here, we focus on the beta-Laguerre ensemble $A_N$ and its joint probability density function (E.1), with $\beta = \beta(N)$ and $D = D(N) \geq N$. For each value of $N$, let $\mu_N$ denote the empirical spectral distribution of $A_N$. Specifically, $\mu_N$ is the measure defined as $\mu_N = N^{-1} \sum_{i=1}^N \delta_{\lambda_i(A_N)}$.

We now present the limits of the empirical spectral distribution of $A_N$, for two distinct regimes of $\beta(N)$ or, equivalently, $\kappa = \kappa(N) = \beta \cdot N$. These have been summarized by Dung & Duy (2021). Later, we will use these results to prove Theorem 4.1.

Firstly, we review the asymptotic results when $N \cdot \beta(N) \to \infty$. In this case, the Marchenko-Pastur distribution appears in the limiting spectral distribution. The Marchenko-Pastur distribution $\mathbf{MP}_\gamma$ with parameter $\gamma \in (0, 1)$ is absolutely continuous on $(0, \infty)$ with finite support only on the interval $I_\gamma = [\gamma_-, \gamma_+]$ where $\gamma_\pm = (1 \pm \sqrt{\gamma})^{1/2}$. The corresponding probability density function is given by

$$\rho_\gamma(x) = \frac{1}{2\pi} \frac{\sqrt{(\gamma_+ - x)(x - \gamma_-)}}{\gamma x}, \qquad x \in I_\gamma. \tag{E.3}$$

**Theorem E.1** (MARCHENKO-PASTUR THEOREM). *If $\beta(N) = \Omega(N^{-1})$ as $N \to \infty$ and $N/D \to \gamma \in (0, 1)$, then the empirical spectral distributions $\mu_N$ of beta-Laguerre ensemble $A_N$, defined by (E.1), converge weakly to the Marchenko-Pastur distribution $\mathbf{MP}_\gamma$ whose probability density function is $\rho_\gamma(x)$ with parameter $\gamma$, defined by (E.3).*

This Marchenko-Pastur Theorem has been established by Dumitriu & Edelman (2006), for any fixed value of $\beta$. Subsequently, Dung & Duy (2021) extended this theorem to encompass general $\beta(N)$, as long as the limit of $N \cdot \beta(N)$ approaches infinity.

Secondly, we consider the high temperature case where $\beta(N) \sim \kappa N^{-1}$ for some fixed $\kappa > 0$. Let $U(a, b, z)$ denote the Tricomi confluent hypergeometric function (DLMF, 13.2.42), defined for negative arguments when $b$ is not a negative

integer by

$$U(a, b, -z) = \frac{\Gamma(1-b)}{\Gamma(a-b+1)} {}_1F_1(a, b, -z) + \frac{\Gamma(b-1)}{\Gamma(a)} z^{1-b} e^{i\pi b} {}_1F_1(a-b+1, 2-b, -z), \qquad z > 0, \qquad \text{(E.4)}$$

where $\Gamma$ is the Gamma function and ${}_pF_q$ is the generalized hypergeometric function. The integer case can be obtained by taking appropriate limits. Below, we present the result of the beta-Laguerre ensemble for the high-temperature case, proved by Dung & Duy (2021) using moment methods.

**Theorem E.2.** *Assume that $\kappa(N) = \beta(N)/N \to \kappa \in (0, \infty)$ as $N \to \infty$ and $N/D \to \gamma \in (0, 1)$. The empirical spectral distributions $\mu_N$ of beta-Laguerre ensemble $A_N$ defined by* (E.1)*, converge weakly to a probability distribution on $(0, \infty)$ with a probability density function $\rho_{\gamma,\kappa}$ defined as*

$$\rho_{\gamma,\kappa}(x) = \frac{\kappa}{2\gamma} \frac{1}{\Gamma(\kappa/2+1)\Gamma(\kappa/2\gamma)} \frac{\left(\frac{\kappa x}{2\gamma}\right)^{\frac{\kappa}{2\gamma}-1-\frac{\kappa}{2}} e^{-\frac{\kappa x}{2\gamma}}}{|U(\kappa/2, -\frac{\kappa}{2\gamma}+1+\frac{\kappa}{2}; -\kappa x/2\gamma)|^2}, \quad x \geq 0. \qquad \text{(E.5)}$$

We call this limiting spectrum the *high-temperature Marchenko-Pastur* (HTMP) distribution, denoted by **HTMP**$_{\gamma,\kappa}$ with parameters $\kappa, \gamma > 0$.

### E.2. Proof of Theorem 4.1

To prove Theorem 4.1, let $M_N$ be an $N \times N$ matrix possessing the high-temperature inverse-Wishart ensemble defined in (6), that is,

$$q_\kappa(\lambda_1, \ldots, \lambda_N) \propto \prod_{i=1}^{N} \lambda_i^{-\alpha} e^{-\beta \lambda_i^{-1}} \prod_{\substack{i,j=1,\ldots,N \\ i<j}} |\lambda_i - \lambda_j|^{\kappa/N}.$$

We focus on the matrix $M_N^{-1}$, whose eigenvalues are reciprocals of those of $M_N$. By a change of variables $\lambda_i \mapsto \lambda_i^{-1}$, the joint eigenvalue density of $M_N^{-1}$, denoted by $\tilde{q}_\kappa$, satisfies

$$\tilde{q}_\kappa(\lambda_1, \ldots, \lambda_N) \propto \prod_{i=1}^{N} \lambda_i^{\alpha-2} e^{-\beta \lambda_i} \prod_{\substack{i,j=1,\ldots,N \\ i<j}} \left| \frac{1}{\lambda_i} - \frac{1}{\lambda_j} \right|^{\kappa/N}.$$

Furthermore, observe that

$$\prod_{\substack{i,j=1,\ldots,N \\ i<j}} \left| \frac{1}{\lambda_i} - \frac{1}{\lambda_j} \right|^{\kappa/N} = \prod_{\substack{i,j=1,\ldots,N \\ i<j}} \left| \frac{\lambda_j - \lambda_i}{\lambda_i \lambda_j} \right|^{\kappa/N}$$

$$= \prod_{i=1,\ldots,N} \lambda_i^{-\kappa+\kappa/N} \prod_{\substack{i,j=1,\ldots,N \\ i<j}} |\lambda_j - \lambda_i|^{\kappa/N},$$

and so

$$\tilde{q}_\kappa(\lambda_1, \ldots, \lambda_N) \propto \prod_{i=1}^{N} \lambda_i^{\alpha-2-\kappa+\kappa/N} e^{-\beta \lambda_i} \prod_{\substack{i,j=1,\ldots,N \\ i<j}} |\lambda_j - \lambda_i|^{\kappa/N}.$$

Another change of variables $\lambda_i \mapsto \frac{2\gamma\beta}{\kappa} \lambda_i$ shows that the matrix $V_N = \frac{2\gamma\beta}{\kappa} M_N^{-1}$ has joint eigenvalue density

$$q(\lambda_1, \ldots, \lambda_N) \propto \prod_{i=1}^{N} \lambda_i^{\alpha-2-\kappa+\kappa/N} e^{-\frac{\kappa}{2\gamma} \lambda_i} \prod_{\substack{i,j=1,\ldots,N \\ i<j}} |\lambda_j - \lambda_i|^{\kappa/N}. \qquad \text{(E.6)}$$

Notice that $\alpha - 2 - \kappa = \frac{\kappa}{2}(\frac{1}{\gamma} - 1) - 1$ since we denote

$$\gamma = \frac{\kappa/2}{\alpha - \kappa/2 - 1},$$

which turns the joint eigenvalue density of $V_N$ into

$$q(\lambda_1, \ldots, \lambda_N) \propto \prod_{i=1}^{N} \left( \lambda_i^{\frac{\kappa}{2}(\frac{1}{\gamma}-1)-1+\frac{\kappa}{N}} e^{-\frac{\kappa}{2\gamma}\lambda_i} \right) \prod_{\substack{i,j=1,\ldots,N \\ i<j}} |\lambda_j - \lambda_i|^{\kappa/N},$$

which is asymptotically equivalent to (E.2) as $N \to \infty$. Hence, it is also asymptotically equivalent to (E.1), where we take $\beta(N) = \kappa(N)/N$ and $N/D \to \gamma \in (0,1)$ in (E.1). Notice that $\gamma \in (0,1)$ indicates that we need to assume $\alpha > \kappa + 1$. Applying Theorems E.1 and E.2 shows convergence in distribution for two different regimes of $\beta(N)$. Therefore, we can conclude that

- when $\kappa(N) \to \infty$, the empirical spectrum distribution of $V_N$ converges weakly to $\mathbf{MP}_\gamma$; and

- when $\kappa(N) \to \kappa$, the empirical spectrum distribution of $V_N$ converges weakly to $\mathbf{HTMP}_{\gamma,\kappa}$.

This completes the proof of Theorem 4.1.

### E.3. Proof of Theorem 4.3

The proof of (8) in Theorem 4.3 is directly based on the Master Model Ansatz (2) and free probability theory. For more details on free probability theory, the definitions of multiplicative free convolution and asymptotic freeness, and their connections with RMT, see Chapter 5 of (Anderson et al., 2009). Precisely, to prove (8), we can apply an independent $N \times N$ unitary matrix $U_N$ following the Haar measure to $\Sigma$. Then, Corollary 5.4.11 in (Anderson et al., 2009) proves that $M_N$ and $U_N \Sigma U_N^*$ are asymptotically free. Thus, we can conclude that the limiting eigenvalue distribution is the multiplicative free convolution $\mu_\Sigma \boxtimes \rho$. The precise definition of the multiplicative free convolution "$\boxtimes$" of two probability distributions can be found in Definition 5.3.28 of Anderson et al. (2009).

Hence, when $\Sigma$ is identity and $M_N$ follows the high-temperature inverse-Wishart ensemble (6), we can apply Theorem 4.1 to conclude that empirical spectrum density $\rho_N(\lambda) \to \rho(\lambda)$ weakly, where the limiting density $\rho(\lambda) = \lambda^{-2}\rho_{\text{HTMP}}(\lambda^{-1})$ if $\kappa$ is finite, and $\rho(\lambda) = \lambda^{-2}\rho_{\text{MP}}(\lambda^{-1})$ if $\kappa = \infty$. Here $\rho_{\text{MP}}$ and $\rho_{\text{HTMP}}$ are the probability density functions of $\mathbf{MP}_\gamma$ and $\mathbf{HTMP}_{\gamma,\kappa}$ defined in Theorems E.1 and E.2, respectively.

Now, we further identify the tail asymptotics of the limiting spectral density $\rho(x)$ for two different cases of $\kappa(N)$.

First, as $\kappa(N) \to \infty$, applying (E.3), we know that the limiting spectral density of $\frac{\kappa}{2\gamma\beta}M_N$ is

$$\rho(x) = \frac{1}{2\pi} \frac{\sqrt{(\gamma_+ - x^{-1})(x^{-1} - \gamma_-)}}{\gamma x}, \qquad x \in [\gamma_+^{-1}, \gamma_-^{-1}],$$

assuming that $\gamma \neq 1$. In this case, the density has bounded support. If $\gamma = 1$, then

$$\rho(x) = \frac{1}{2\pi} \frac{\sqrt{2x^{-1} - x^{-2}}}{x}, \qquad x \in [2, \infty),$$

and so $\rho(x) \sim \frac{1}{\pi\sqrt{2}} x^{-3/2}$ as $x \to \infty$. Scaling these densities does not influence the tail behavior.

Next, we turn to the case $\kappa(N) \to \kappa \in (0,\infty)$ as $N \to \infty$, and we rely on two asymptotic properties of the Tricomi confluent hypergeometric function defined in (E.4):

(I) For any $b < 0$, $U(a,b,z) = \frac{\Gamma(1-b)}{\Gamma(a-b+1)} + \mathcal{O}(z)$ as $|z| \to 0$ (DLMF, 13.2.22); and

(II) For any $a, b$, $|U(a,b,z)| \sim |z|^{-a}$ as $|z| \to \infty$ with $|\arg z| < \frac{3\pi}{2}$ (DLMF, 13.2.6).

From (I) and (II), the probability density function $\rho_{\gamma,\kappa}(x)$ given in (E.5) satisfies

$$\rho_{\gamma,\kappa}(x) \sim c_1 x^{\frac{\kappa}{2\gamma}-1-\frac{\kappa}{2}} \text{ as } x \to 0^+, \quad \text{and} \quad \rho_{\gamma,\kappa}(x) \sim c_2 x^{\frac{\kappa}{2\gamma}-1+\frac{\kappa}{2}} e^{-\frac{\kappa}{2\gamma}x} \text{ as } x \to \infty.$$

Thus, with a change of variables, the density function $\rho$ of $\frac{\kappa}{2\gamma\beta}M_N^{-1}$ satisfies

$$\rho(x) \sim c_1 x^{-\frac{\kappa}{2\gamma}-1+\frac{\kappa}{2}} \text{ as } x \to \infty, \quad \text{and} \quad \rho(x) \sim c_2 x^{-\frac{\kappa}{2\gamma}-1-\frac{\kappa}{2}} \exp\left(-\frac{\kappa}{2\gamma x}\right) \text{ as } x \to 0^+.$$

We now only need to observe that rescaling $M_N$ will not change the exponent in the power law, but it will change the coefficient in the exponential term.

### E.4. Properties of High-Temperature Marchenko-Pastur Distribution

Recall that the Stieltjes transform of a spectral density $\rho$ is given by $S_\rho(z) = \int (x-z)^{-1}\rho(x)\mathrm{d}x$. Thus, we have the following result.

**Proposition E.3.** *The Stieltjes transform of* $\mathbf{HTMP}_{\gamma,\kappa}$ *defined in* (E.5) *is given by*

$$S_{\rho_{\gamma,\kappa}}(z) = \int_0^\infty \frac{\rho_{\gamma,\kappa}(x)}{x-z}\mathrm{d}x = \frac{\kappa}{2\gamma} \cdot \frac{U(\kappa/2+1, 2-\frac{\kappa}{2\gamma}+\frac{\kappa}{2}; -\frac{\kappa z}{2\gamma})}{U(\kappa/2, -\frac{\kappa}{2\gamma}+1+\frac{\kappa}{2}; -\frac{\kappa z}{2\gamma})}, \quad z \in \mathbb{C} \setminus \mathbb{R}. \tag{E.7}$$

The proof of the above proposition is given by Ismail et al. (1988).

From DLMF, 13.7.3, the asymptotic expansion for large $z$ of $U(a,b,z)$ is given by

$$U(a,b,z) \sim z^{-a} \sum_{k=0}^\infty c_k^{(a,b)} z^{-k}, \qquad c_k^{(a,b)} := (-1)^k \frac{\Gamma(a+k)\Gamma(a-b+k+1)}{k!\Gamma(a)\Gamma(a-b+1)}. \tag{E.8}$$

Using (E.7) and this approximation of $U(a,b,z)$, we can compute the first few moments of the $\mathbf{HTMP}_{\gamma,\kappa}$ distribution. These are given in the following proposition.

**Proposition E.4.** *Let* $m_k = \int_0^\infty x^k \rho_{\gamma,\kappa}(x)\mathrm{d}x$ *denote the $k$-th moment of the* $\mathbf{HTMP}_{\gamma,\kappa}$ *distribution for any* $k \in \mathbb{N}$. *Then* $m_0 = 1$, $m_1 = -(a-b+1)$, *and* $m_2 = (a-b+1)(2a-b+2)$ *where* $a = \frac{\kappa}{2}$ *and* $b = 1 + \frac{\kappa}{2} - \frac{\kappa}{2\gamma}$. *In general,*

$$m_k = \frac{k}{a}c_k^{(a,b)} - \sum_{l=1}^{k-1} m_{k-l}c_l^{(a,b)},$$

*where* $c_k^{(a,b)}$ *is defined in* (E.8).

*Proof.* We seek an expansion of the form $S_{\rho_{\gamma,\kappa}}(z) = \sum_{k=0}^\infty m_k z^{-k-1}$, as the coefficients $m_k$ are precisely the moments of $\mathbf{HTMP}_{\gamma,\kappa}$. Let $a = \frac{\kappa}{2}$ and $b = 1 + \frac{\kappa}{2} - \frac{\kappa}{2\gamma}$. From (E.7) and (E.8), formally,

$$\sum_{k=0}^\infty m_k z^{-k-1} = \frac{z^{-a-1} \sum_{k=0}^\infty c_k^{(a+1,b+1)} z^{-k}}{z^{-a} \sum_{k=0}^\infty c_k^{(a,b)} z^{-k}},$$

and so

$$\left(\sum_{k=0}^\infty m_k z^{-k}\right)\left(\sum_{k=0}^\infty c_k^{(a,b)} z^{-k}\right) = \sum_{k=0}^\infty c_k^{(a+1,b+1)} z^{-k}.$$

Using the Cauchy product,

$$\sum_{k=0}^\infty \left(\sum_{l=0}^k m_{k-l}c_l^{(a,b)}\right) z^{-k} = \sum_{k=0}^\infty c_k^{(a+1,b+1)} z^{-k},$$

and so for $k \geq 0$,

$$m_k + \sum_{l=1}^k m_{k-l}c_l^{(a,b)} = c_k^{(a+1,b+1)}.$$

Evaluating $m_k$ for $k \in \{0,1\}$, we find that $m_0 = 1$ (as expected),

$$m_1 = -(a-b+1),$$

where we use the fact that $\Gamma(z + 1) = z\Gamma(z)$. Furthermore, for $k = 2$, we have

$$m_2 = (a - b + 1)(2a - b + 2).$$

$\square$

## F. Structured Matrix Models: Weyl Formulae and Joint Eigenvalue Densities

In this section, we derive a change-of-variable formulae (also called Weyl formulae) for obtaining joint eigenvalue densities with structured matrix models. The models we consider are outlined in Table F.1. The results proved in this section will be applied later in Appendix G to solve the variational problem (7) for various matrix models outlined in Table F.1.

| Type | Description | Weyl Formula | General Form |
|---|---|---|---|
| (a) *Diagonal* | All off-diagonal entries are zero | Trivial | $A = \mathrm{diag}(\lambda_1, \ldots, \lambda_n)$ |
| (b) *Commuting block diag.* | Block-diagonal with symmetric blocks $A_i = A_i^\top$, plus $[A_i, A_j] = 0$ for all $i, j$ | Thm. F.2 | $A = \mathrm{diag}(A_1, \ldots, A_k)$, where $[A_i, A_j] = 0$ |
| (c) *Symmetric block diag.* | Block-diagonal with symmetric blocks $A_i = A_i^\top$ | Cor. F.4 | $A = \mathrm{diag}(A_1, \ldots, A_k)$ |
| (d) *Kronecker-like matrix* | Full symmetric matrix with commuting subblocks | Thm. F.5 | $A = [A_{ij}]_{i,j=1}^k$, where $A_{ij} = A_{ji}^\top$ and $[A_{ij}, A_{kl}] = 0$ |
| (e) *Symmetric* | General symmetric matrix | Cor. F.3 | $A = A^\top$ |

*Table F.1.* A collection of structured matrix models with explicit joint eigenvalue densities.

Computing Jacobians for matrix-valued functions presents a notationally complex task. To address this challenge, we employ the following technique derived from the change of coordinates relationships prevalent in Riemannian geometry, as outlined in (Menon & Trogdon, 2015).

> **Change of Coordinates Trick:** Assuming that $M = F(x_1, \ldots, x_N)$ is a matrix-valued function, let $\partial M = (\partial M)_{ij}$ where $(\partial M)_{ij} = \sum_{k=1}^N J_{ijk} \mathrm{d}x_k$ and $J_{ijk} = \frac{\partial F_{ij}}{\partial x_k}$ are the elements of the corresponding Jacobian matrix. Compute
>
> $$\mathrm{tr}(\partial M^\top \partial M) = \sum_{k,l=1}^N g_{kl} \mathrm{d}x_k \mathrm{d}x_l \quad \text{where} \quad g_{kl} = \sum_{i,j} J_{ijk} J_{ijl}.$$
>
> Then for any integrable function $f$,
>
> $$\int f(M)\mathrm{d}M = \int f(F(x_1, \ldots, x_N))\sqrt{\det g(x_1, \ldots, x_N)} \, \mathrm{d}x_1 \cdots \mathrm{d}x_N.$$

As a simple example of this trick, Lemma F.1 considers the task of finding the Haar measure (the uniform distribution) on the space of symmetric matrices.

**Lemma F.1** (SYMMETRIC MATRICES). *The Haar measure on the group of $N \times N$ real-valued symmetric matrices is given by*

$$\mathrm{d}M = 2^{\frac{N(N-1)}{4}} \prod_{j=1}^N \mathrm{d}M_{jj} \prod_{1 \le j < k \le N} \mathrm{d}M_{jk}.$$

*Proof.* Since $\mathrm{d}M_{kj} = \mathrm{d}M_{jk}$, it follows that

$$\mathrm{tr}(\partial M^\top \partial M) = \sum_{j,k} \mathrm{d}M_{jk} \mathrm{d}M_{kj} = \sum_k \mathrm{d}M_{kk}^2 + 2 \sum_{j<k} \mathrm{d}M_{jk}^2.$$

Therefore, for $M \in \mathbb{R}^{N \times N}$, $g_{ii} = 1$ and $g_{ij} = 2$ for $i \neq j$. Hence, $\det g = 1^N \cdot 2^{\frac{N(N-1)}{2}} = 2^{\frac{N(N-1)}{2}}$, and the result follows. $\qquad\square$

We are now interested in applying the trick to obtain joint densities of eigenvalues starting from densities over Haar measure on subclasses of symmetric matrices. The most famous result in this regard is the Weyl formula for the joint density of eigenvalues over the general class of symmetric matrices. This is presented in Corollary F.3, as a special case of a useful generalization that we present next in Theorem F.2. This next result treats our commuting block diagonal matrices (type (b)), and is inspired by McCarthy (2023), involving $d$ real-valued symmetric matrices that are simultaneously-diagonalizable. That is, these are families of matrices $M_1, \ldots, M_n$ such that $M_i = Q\Lambda_i Q^\top$, where $Q$ is orthogonal and each $\Lambda_i$ is diagonal. Note that if $N = mn$, then $N \times N$ matrices which are block-diagonal with commuting $m \times m$ blocks can be treated in this way.

**Theorem F.2** (WEYL FORMULA: TYPE (B)). *Let* $\mathrm{d}M$ *be the Haar measure on the space* $\mathcal{M}_m^n$ *of $n$-simultaneously-diagonalizable matrices of size $m \times m$. Let* $f : \mathcal{M}_m^n \to \mathbb{R}$ *be such that $f$ satisfies $f(QMQ^\top) = f(M)$ where $Q \in \mathbb{R}^{m \times m}$ is orthogonal and $M \in \mathcal{M}_m^n$. Then, there exists a constant $C_{m,n} > 0$ independent of $f$ such that*

$$\int f(M)\mathrm{d}M = C_{m,n} \int f(\Lambda) \prod_{\substack{i,j=1,\ldots,m \\ i<j}} \|\lambda_{.j} - \lambda_{.i}\| \mathrm{d}\Lambda,$$

*where* $\Lambda = (\Lambda_1, \ldots, \Lambda_n)$ *and* $\Lambda_i = \mathrm{diag}\{\lambda_{i1}, \ldots, \lambda_{im}\}$.

*Proof.* Immediately, we restrict the support to those matrices with distinct eigenvalues. One can readily verify that the set of matrices with non-distinct eigenvalues has zero measure with respect to the Haar measure on symmetric matrices, so this does not affect the density. Let $\Lambda = (\Lambda_1, \ldots, \Lambda_n)$, so that $M = Q(\Lambda_1, \ldots, \Lambda_n)Q^\top$, where $M = (M_1, \ldots, M_n)$. Then

$$\partial M = \mathrm{d}Q(\Lambda_1, \ldots, \Lambda_n)Q^\top + Q(\mathrm{d}\Lambda_1, \ldots, \mathrm{d}\Lambda_n)Q^\top + Q(\Lambda_1, \ldots, \Lambda_n)\mathrm{d}Q^\top$$
$$= Q\left[(\mathrm{d}\Lambda_1, \ldots, \mathrm{d}\Lambda_n) + Q^\top \mathrm{d}Q(\Lambda_1, \ldots, \Lambda_n) + (\Lambda_1, \ldots, \Lambda_n)\mathrm{d}Q^\top Q\right]Q^\top.$$

Since $Q^\top Q = I$, this implies that $\mathrm{d}Q^\top Q = -Q^\top \mathrm{d}Q$, and so

$$\partial M = Q\left[(\mathrm{d}\Lambda_1, \ldots, \mathrm{d}\Lambda_n) + Q^\top \mathrm{d}Q(\Lambda_1, \ldots, \Lambda_n) - (\Lambda_1, \ldots, \Lambda_n)Q^\top \mathrm{d}Q\right]Q^\top$$
$$= Q\left[\mathrm{d}\Lambda + [Q^\top \mathrm{d}Q, \Lambda]\right]Q^\top.$$

Consider the variable $A$ such that $\mathrm{d}A = Q^\top \mathrm{d}Q$, so that $\partial M = Q(\mathrm{d}\Lambda + [\mathrm{d}A, \Lambda])Q^\top$. Now,

$$\mathrm{tr}(\partial M^\top \partial M) = \sum_{i=1}^n \mathrm{tr}(\mathrm{d}\Lambda_i^2) + 2\mathrm{tr}(\mathrm{d}\Lambda_i[\mathrm{d}A, \Lambda_i]) + \mathrm{tr}[\mathrm{d}A, \Lambda_i]^2.$$

Observe that for a diagonal matrix $P = \mathrm{diag}\{p_1, \ldots, p_m\}$, we have that

$$[\mathrm{d}A, P]_{ij} = (\mathrm{d}AP - P\mathrm{d}A)_{ij} = \mathrm{d}A_{ij}p_j - p_i\mathrm{d}A_{ij} = (p_j - p_i)\mathrm{d}A_{ij},$$

and so

$$\begin{aligned}
\mathrm{tr}([\mathrm{d}A, P])^2 &= \sum_{i,j=1}^m [\mathrm{d}A, P]_{ij}[\mathrm{d}A, P]_{ji} \\
&= \sum_{i,j=1}^m (p_j - p_i)\mathrm{d}A_{ij}(p_i - p_j)\mathrm{d}A_{ji} \\
&= \sum_{i,j=1}^m (p_j - p_i)^2\mathrm{d}A_{ij}^2, \\
&= 2\sum_{i<j} (p_j - p_i)^2\mathrm{d}A_{ij}^2,
\end{aligned}$$

where the last two lines follow from the fact that $\mathrm{d}A = \mathrm{d}Q^\top Q$ is anti-symmetric, and so $(\mathrm{d}A)_{ii} = 0$ and $(\mathrm{d}A)_{ik} = -(\mathrm{d}A)_{ki}$. Similarly,

$$\mathrm{tr}(\mathrm{d}P[\mathrm{d}A, P]) = \sum_k \mathrm{d}p_k[\mathrm{d}A, P]_{kk} = 0,$$

since $[\mathrm{d}A, P]_{kk} = 0$. Consequently,

$$\mathrm{tr}(\partial M^\top \partial M) = \sum_{i=1}^n \sum_{j=1}^m \mathrm{d}\lambda_{ij}^2 + 2\sum_{j<k} \|\lambda_{\cdot j} - \lambda_{\cdot k}\|^2 \mathrm{d}A_{jk}^2. \tag{F.1}$$

Therefore, using the change of coordinates trick,

$$\int f(M)\mathrm{d}M = 2^{\frac{m(m-1)}{2}} \int f(\Lambda, A) \prod_{j<k} \|\lambda_{\cdot j} - \lambda_{\cdot k}\| \prod_{i=1}^n \prod_{j=1}^m \mathrm{d}\lambda_{ij}^2 \prod_{j<k} \mathrm{d}A_{jk}^2,$$

and for $f$ independent of $A$, the integral over the measure of $A$ separates, and so for some $C_{m,n} > 0$ independent of $f$, we have that

$$\int f(M)\mathrm{d}M = C_{m,n} \int f(\Lambda) \prod_{j<k} \|\lambda_{\cdot j} - \lambda_{\cdot k}\|^2 \mathrm{d}\Lambda.$$

$\square$

From this result, we immediately obtain the following two results as corollaries.

**Corollary F.3** (WEYL FORMULA: TYPE (E))**.** *Let* $\mathrm{d}M$ *be the Haar measure on the space of real-valued symmetric matrices of size* $N \times N$*. Let* $f$ *be a matrix function that depends only on the eigenvalues. Then there exists a constant* $C_N > 0$ *independent of* $f$ *such that*

$$\int f(M)\mathrm{d}M = C_N \int f(\lambda_1, \ldots, \lambda_N) \prod_{\substack{j,k=1,\ldots,N \\ j<k}} |\lambda_j - \lambda_k| \prod_{i=1}^N \mathrm{d}\lambda_i.$$

**Corollary F.4** (WEYL FORMULA: TYPE (C))**.** *Let* $\mathrm{d}M$ *be the Haar measure on the space of real-valued symmetric matrices of size* $N \times N$ *with* $N = mn$*, that are block-diagonal with block size* $m \times m$*. Let* $f$ *be a matrix function that depends only on the eigenvalues. Then there exists a constant* $C_{m,n} > 0$ *independent of* $f$ *such that*

$$\int f(M)\mathrm{d}M = C_{m,n} \int f(\Lambda) \prod_{i=1}^n \prod_{\substack{j,k=1,\ldots,m \\ j<k}} |\lambda_{ij} - \lambda_{ik}| \mathrm{d}\Lambda.$$

Finally, we consider the case of general block matrices with commuting blocks. This case is slightly different than the setting in Theorem F.2, as the orthogonal decomposition adopts a Kronecker-like structure. However, we can use the same techniques to obtain the joint eigenvalue density.

**Theorem F.5** (WEYL FORMULA: TYPE (D))**.** *Let* $\mathrm{d}M$ *be the Haar measure on the space* $\mathcal{M}_{m,n}$ *of* $mn \times mn$ *real-valued symmetric matrices comprised of* $n \times n$ *commuting* $m \times m$ *blocks. Let* $f : \mathcal{M}_{m,n} \to \mathbb{R}$ *be such that* $f$ *satisfies* $f(QMQ^\top) = f(M)$*, where* $Q \in \mathbb{R}^{m \times m}$ *is orthogonal and* $M \in \mathcal{M}_{m,n}$*. Then there exists a constant* $C_{m,n} > 0$ *independent of* $f$ *such that*

$$\int f(M)\mathrm{d}M = C_{m,n} \int f(\Lambda) \prod_{\substack{i,j=1,\ldots,m \\ i<j}} \left( \sum_{k,l=1}^n (\lambda_{jl} - \lambda_{ik})^2 \right)^{1/2} \prod_{\substack{i,j=1,\ldots,n \\ i<j}} \left( \sum_{k,l=1}^m (\lambda_{lj} - \lambda_{ki})^2 \right)^{1/2} \mathrm{d}\Lambda,$$

*where* $\Lambda = (\Lambda_1, \ldots, \Lambda_n)$ *and* $\Lambda_i = \mathrm{diag}\{\lambda_{i1}, \ldots, \lambda_{im}\}$*.*

*Proof.* The proof proceeds in a similar fashion to the proof of Theorem F.2. Once again, the support of the density is appropriately restricted, without loss of generality. Now consider the change of coordinates

$$M = (Q \otimes O)^\top \mathrm{diag}\{\Lambda_1, \ldots, \Lambda_n\}(Q \otimes O),$$

where $Q \in \mathbb{R}^{n \times n}$ and $O \in \mathbb{R}^{m \times m}$ are both orthogonal, and so $U = Q \otimes O$ is also orthogonal. Note that $\mathrm{d}U = (\mathrm{d}Q \otimes O) + (Q \otimes \mathrm{d}O)$. As before,

$$\begin{aligned}
\partial M &= U(\mathrm{d}\Lambda + [U^\top \mathrm{d}U, \Lambda])U^\top \\
&= U(\mathrm{d}\Lambda + [(Q \otimes O)^\top(\mathrm{d}Q \otimes O), \Lambda] + [(Q \otimes O)^\top(Q \otimes \mathrm{d}O), \Lambda])U^\top \\
&= U(\mathrm{d}\Lambda + [(Q^\top \mathrm{d}Q \otimes I), \Lambda] + [(I \otimes O^\top \mathrm{d}O), \Lambda])U^\top.
\end{aligned}$$

Now, consider variables $A$ and $B$ such that $\mathrm{d}A = Q^\top \mathrm{d}Q$ and $\mathrm{d}B = O^\top \mathrm{d}O$. Since $\mathrm{d}A$ and $\mathrm{d}B$ are both antisymmetric, for $\mathrm{d}A \oplus \mathrm{d}B = (\mathrm{d}A \otimes I) + (I \otimes \mathrm{d}B)$, we have that

$$(\mathrm{d}A \oplus \mathrm{d}B)^\top = (\mathrm{d}A^\top \otimes I) + (I \otimes \mathrm{d}B^\top) = -(\mathrm{d}A \otimes I) - (I \otimes \mathrm{d}B) = -(\mathrm{d}A \oplus \mathrm{d}B),$$

and it is therefore also antisymmetric. Recall that

$$\begin{aligned}
(\mathrm{d}A \otimes I)_{m(i-1)+k, m(j-1)+l} &= \mathrm{d}A_{ij} \\
(I \otimes \mathrm{d}B)_{m(i-1)+k, m(j-1)+l} &= \mathrm{d}B_{kl},
\end{aligned}$$

and so

$$\begin{aligned}
[(\mathrm{d}A \otimes I), \Lambda]_{m(i-1)+k, m(j-1)+l} &= (\lambda_{jl} - \lambda_{ik})\mathrm{d}A_{ij} \\
[(I \otimes \mathrm{d}B), \Lambda]_{m(i-1)+k, m(j-1)+l} &= (\lambda_{jl} - \lambda_{ik})\mathrm{d}B_{kl}.
\end{aligned}$$

Consequently,

$$\mathrm{tr}([(\mathrm{d}A \otimes I), \Lambda]^2) = 2 \sum_{i<j} \left( \sum_{k,l} (\lambda_{jl} - \lambda_{ik})^2 \right) \mathrm{d}A_{ij}^2$$

$$\mathrm{tr}([(I \otimes \mathrm{d}B), \Lambda]^2) = 2 \sum_{k<l} \left( \sum_{i,j} (\lambda_{jl} - \lambda_{ik})^2 \right) \mathrm{d}B_{kl}^2.$$

On the other hand, since $\mathrm{d}A$ and $\mathrm{d}B$ are antisymmetric,

$$\mathrm{tr}([(\mathrm{d}A \otimes I), \Lambda][(I \otimes \mathrm{d}B), \Lambda]) = \sum_{i,j,k,l} (\lambda_{jl} - \lambda_{ik})^2 \mathrm{d}A_{ij}\mathrm{d}B_{kl} = 0.$$

Altogether,

$$\mathrm{tr}(\partial M^\top \partial M) = \sum_{i=1}^{n} \sum_{j=1}^{m} \mathrm{d}\lambda_{ij}^2 + 2 \sum_{i<j} \left( \sum_{k,l} (\lambda_{jl} - \lambda_{ik})^2 \right) \mathrm{d}A_{ij}^2 + 2 \sum_{k<l} \left( \sum_{i,j} (\lambda_{jl} - \lambda_{ik})^2 \right) \mathrm{d}B_{kl}^2. \tag{F.2}$$

The result follows by the change of coordinates trick. $\qquad\square$

*Remark* F.6. In each case, the Weyl change-of-variables factor in the joint eigenvalue densities involves absolute differences of eigenvalues $|\lambda_i - \lambda_j|$, and it varies only in how these differences are included. These factors determine the degree of *eigenvalue repulsion*, as they assert that configurations with eigenvalues close together should occur with reduced probability. *This plays a central role in the shape of the spectral density.* An important observation for us is that the strength of these repulsions only increases as we take products of eigenvalue differences, since sums do not further reduce the probability density. Heuristically, we can count the number of multiplied eigenvalue differences as a rough approximation of the degree of eigenvalue repulsion in the spectral density. Let us call this quantity $\theta$. From the Weyl formulae, for each structured matrix class, $\theta$ is as follows.

(a) *Diagonal:* $\theta = 0$

(b) *Commuting block diagonal:* $\theta = \frac{m(m-1)}{2}$

(c) *Symmetric block diagonal:* $\theta = \frac{N(m-1)}{2}$

(d) *Kronecker-like matrix:* $\theta = \frac{m(m-1)}{2} + \frac{n(n-1)}{2}$

(e) *Symmetric:* $\theta = \frac{N(N-1)}{2}$

Clearly, $\theta$ becomes larger as the matrix class becomes less structured (more degrees of freedom).

In principle, one can measure the amount of matrix structure according to $\theta$. Finding the spectral density of structured matrices from their joint eigenvalue densities can be very challenging, but this becomes much simpler under an approximation that enforces all eigenvalues to be *interchangeable*, that is, when the joint density is symmetric. Letting $w(\lambda_1, \ldots, \lambda_N)$ denote the Weyl change-of-variables factor, we can approximate $w$ using Schur functions (Noumi, 2023, §3). This process was recently proposed in the context of RMT in Kimura & Mazenc (2021). To reduce complexity, we can further observe that the behavior of $w$ will be dominated by the leading-order term $\prod_{i<j} |\lambda_i - \lambda_j|^\gamma$ for some $\gamma$ (Noumi, 2023, pp. 21). Therefore, a class of approximations that should respect the degree of eigenvalue repulsion, as measured by $\theta$, weights each eigenvalue difference according to the ratio $\theta / \frac{N(N-1)}{2}$, so that

$$\prod_{i,j \in E} |\lambda_i - \lambda_j| \qquad \text{becomes} \qquad \prod_{\substack{i,j=1,\ldots,n \\ i<j}} |\lambda_i - \lambda_j|^{\frac{2\theta}{N(N-1)}}.$$

This is only a heuristic, however, and this principle must be made more precise. In Appendix G, we show that variational approximations using beta-ensembles replicate this behavior in a principled way.

For completeness, we consider one last example of a Weyl formula which will help to make the comments in Remark F.6 more explicit. Let $M$ be a symmetric matrix with eigendecomposition $Q\Lambda Q^\top$. Here, we consider the scenario where only $d$ columns of $Q$ are variable, and the other $N - d$ are fixed orthogonal vectors. We refer to this scenario as the "$d$ free eigenvectors" case, as only $d$ of the eigenvectors of $M$ are allowed to vary in the random matrix ensemble.

**Corollary F.7** (WEYL FORMULA: FREE EIGENVECTORS). *Let $\mathrm{d}M$ be the Haar measure on the space of real-valued symmetric matrices of size $N \times N$ with $d$ free eigenvectors. Let $f$ be a matrix function that depends only on the eigenvalues. Then there exists a constant $C_{N,d} > 0$ independent of $f$ such that*

$$\int f(M)\mathrm{d}M = C_{N,d} \int f(\lambda_1, \ldots, \lambda_N) \prod_{\substack{j,k=1,\ldots,d \\ j<k}} |\lambda_j - \lambda_k| \prod_{i=1}^N \mathrm{d}\lambda_i.$$

*Proof.* Let $Q = (U|V)$ where $U \in \mathbb{R}^{N \times d}$ is variable and $V \in \mathbb{R}^{N \times (N-d)}$ is fixed. Then

$$Q^\top Q = \begin{pmatrix} U^\top U & U^\top V \\ V^\top U & V^\top V \end{pmatrix} = I,$$

and so

$$\mathrm{d}(Q^\top Q) = \begin{pmatrix} \mathrm{d}U^\top U + U^\top \mathrm{d}U & \mathrm{d}U^\top V \\ V^\top \mathrm{d}U & 0 \end{pmatrix} = 0.$$

This implies that $V^\top \mathrm{d}U = \mathrm{d}U^\top V = 0$. Consequently, since $\mathrm{d}Q = (\mathrm{d}U|0)$,

$$\mathrm{d}A = Q^\top \mathrm{d}Q = \begin{pmatrix} U^\top \mathrm{d}U & 0 \\ V^\top \mathrm{d}U & 0 \end{pmatrix} = \begin{pmatrix} U^\top \mathrm{d}U & 0 \\ 0 & 0 \end{pmatrix}.$$

This implies that $\mathrm{d}A_{ij} = 0$ whenever $i > d$ or $j > d$. From (F.1),

$$\mathrm{tr}(\partial M^\top \partial M) = \sum_{i=1}^N \mathrm{d}\lambda_i^2 + 2 \sum_{\substack{j,k=1,\ldots,d \\ j<k}} |\lambda_j - \lambda_k|^2 \mathrm{d}A_{jk}^2,$$

and by using the change of coordinates trick, the result follows. □

Returning to Remark F.6, we can count the number of eigenvalue differences in Corollary F.7 and find $\theta = \frac{d(d-1)}{2}$. This provides a direct relationship between the number of "degrees of freedom" in the matrix structure according to the number of free eigenvectors, and the resulting Weyl formulae. We will return to this example in Appendix G.

## G. Variational Approximations with Beta-Ensembles

In this section, we use the Weyl formulae from Appendix F to solve the variational problem (7) and to derive the relationships in Table 2 for the matrix models outlined in Table F.1. Our derivations begin in Section G.1 with cases (a), (c), and (e) using an exact formula when the change-of-variables factor is of a particular form. To cover cases (b) and (d) (using (c) only to test the validity of our approach), we estimate the relationships using symbolic regression applied to estimated solutions of (7). This approach is centered around a numerical method outlined and justified in Section G.2. The experiments are performed and conclusions drawn in Section G.3.

Assume that a matrix $M \in \mathbb{R}^{N \times N}$ has elements distributed according to a density $p(M) \propto \prod_{i=1}^{N} e^{-V(\lambda_i)}$ over some class of matrices $\mathcal{M}$, where $\lambda_1, \ldots, \lambda_N$ are the eigenvalues of $M$. Such densities typically arise in the form $p(M) \propto (\det g(M))^{\alpha} e^{-\mathrm{tr} f(M)}$ for some analytic functions $f, g$. After performing a change of variables, the corresponding joint density of eigenvalues adopts the form

$$p(\lambda_1, \ldots, \lambda_N) \propto w(\lambda_1, \ldots, \lambda_N) \prod_{i=1}^{N} e^{-V(\lambda_i)}, \tag{G.1}$$

where $w$ depends on the underlying class of matrices $\mathcal{M}$. To provide a concrete baseline for the study of models of this form, we consider approximation the joint density of eigenvalues by the Laguerre beta-ensemble

$$q_\beta(\lambda_1, \ldots, \lambda_N) = \frac{1}{Z(\beta)} \prod_{i=1}^{N} e^{-V(\lambda_i)} \prod_{\substack{i,j=1,\ldots,N \\ i<j}} |\lambda_i - \lambda_j|^\beta. \tag{G.2}$$

Note that $\beta = 1$ corresponds to the case where $\mathcal{M}$ is the set of real symmetric matrices, and $\beta = 2$ corresponds to the case where $\mathcal{M}$ is the set of complex Hermitian matrices. Intuitively, smaller values of $\beta$ correspond to a more restrictive class $\mathcal{M}$. Tridiagonal matrix models exhibiting joint eigenvalue densities given by (G.2) were discovered in Dumitriu & Edelman (2002). To approximate (G.1) by the variational family (G.2), we consider the forward variational approximation:

$$\beta^* = \operatorname*{argmin}_{\beta \geq 0} d_{\mathrm{KL}}(q_\beta \| p), \qquad \kappa^* = N\beta^*. \tag{G.3}$$

### G.1. Counting Eigenvalue Differences

We shall now confirm that the variational approximation (G.3) replicates the desired heuristic behavior outlined in Remark F.6. In Proposition G.1, we show that when the change of variable factor $w$ is of product form, then $\beta^*$ is explicitly determined by counting the number of eigenvalue differences.

**Proposition G.1.** *For a probability density $p$ satisfying (G.1) with $w(\lambda) = \prod_{ij \in E} |\lambda_i - \lambda_j|^{\theta_{ij}}$, letting $\theta = \sum_{ij \in E} \theta_{ij}$,*

$$\beta^* = \operatorname*{argmin}_{\beta \geq 0} d_{\mathrm{KL}}(q_\beta \| p) = \frac{2\theta}{N(N-1)}.$$

*Proof.* A primary tool in the proof is the explicit expressions for the derivatives of $F(\beta) = \log Z(\beta)$, where

$$Z(\beta) = \int \exp\left( -\sum_{i=1}^{N} V(\lambda_i) + \beta \sum_{i<j} \log |\lambda_i - \lambda_j| \right) d\lambda,$$

from it follows immediately that

$$F'(\beta) = \frac{Z'(\beta)}{Z(\beta)} = \mathbb{E}_{q_\beta} \sum_{i<j} \log |\lambda_i - \lambda_j|$$

$$F''(\beta) = \frac{Z''(\beta)}{Z(\beta)} - \left( \frac{Z'(\beta)}{Z(\beta)} \right)^2 = \operatorname{Var}_{q_\beta} \sum_{i<j} \log |\lambda_i - \lambda_j| > 0.$$

Note that since $q_\beta$ is symmetric in its arguments, each eigenvalue is an exchangeable random variable, and so

$$\mathbb{E}_{q_\beta} \sum_{ij \in E} \theta_{ij} \log |\lambda_i - \lambda_j| = \sum_{ij \in E} \theta_{ij} \mathbb{E}_{q_\beta} \log |\lambda_i - \lambda_j| = \mathbb{E}_{q_\beta} \log |\lambda_1 - \lambda_2| \sum_{ij \in E} \theta_{ij} = \frac{2\theta}{N(N-1)} F'(\beta).$$

On the other hand,

$$\mathbb{E}_{q_\beta} \sum_{i<j} \beta \log |\lambda_i - \lambda_j| = \beta F'(\beta).$$

Altogether, letting $Z_p$ denote the normalizing constant for $p$,

$$d_{\mathrm{KL}}(q_\beta \| p) = \mathbb{E}_{q_\beta} \log \left( \frac{q_\beta(\lambda_1, \ldots, \lambda_N)}{p(\lambda_1, \ldots, \lambda_N)} \right) = \log Z_p + \left( \beta - \frac{2\theta}{N(N-1)} \right) F'(\beta) - F(\beta).$$

Taking the derivative, we obtain

$$\frac{\mathrm{d}}{\mathrm{d}\beta} d_{\mathrm{KL}}(q_\beta \| p) = \left( \beta - \frac{2\theta}{N(N-1)} \right) F''(\beta),$$

and since $F''(\beta) > 0$, the only critical point of $\beta \mapsto d_{\mathrm{KL}}(q_\beta \| p)$ is $\beta = \frac{2\theta}{N(N-1)}$. To show this is a minimizer, observe that

$$\frac{\mathrm{d}^2}{\mathrm{d}\beta^2} d_{\mathrm{KL}}(q_\beta \| p) = F''(\beta) + \left( \beta - \frac{2\theta}{N(N-1)} \right) F'''(\beta),$$

and at the critical point, $\frac{\mathrm{d}^2}{\mathrm{d}\beta^2} d_{\mathrm{KL}}(q_\beta \| p) = F''(\beta) > 0$. $\qquad \square$

From this, we obtain the following result as a corollary.

**Corollary G.2.** *For the following structured matrix models, the values of $\beta^*$, $\kappa^*$ defined in (G.3) are given by:*

- *(a) Diagonal: $\beta^* = 0$, $\kappa^* = 0$.*

- *(c) Symmetric block diagonal: $\beta^* = \frac{m-1}{N-1}$, $\kappa^* = (m-1) \cdot \frac{N}{N-1}$.*

- *(e) Symmetric: $\beta^* = 1$, $\kappa^* = N$.*

*Proof.* The diagonal and symmetric cases are trivial. For the setting of block-diagonal matrices with

$$w(\lambda_1, \ldots, \lambda_N) = \prod_{i=1}^{n} \prod_{\substack{j,k=1,\ldots,m \\ j<k}} |\lambda_{ij} - \lambda_{ik}|,$$

as shown in Corollary F.4, $\theta = n \cdot \frac{m(m-1)}{2}$, and so Proposition G.1 implies $\beta^* = \frac{nm(m-1)}{nm(nm-1)} = \mathcal{O}(n^{-1})$. $\qquad \square$

Performing a similar analysis for our free eigenvectors scenario in Appendix F, we obtain Theorem G.3.

**Theorem G.3.** *Consider the density (2) with $\pi$ satisfying an inverse-Wishart distribution over the space of symmetric matrices with $d$ free eigenvectors. The variational approximation (7) satisfies $\kappa^* = d \cdot \frac{d-1}{N-1} \sim \frac{d^2}{N}$ as $d, N \to \infty$.*

*Proof of Theorem G.3.* Applying Proposition G.1 to Corollary F.4, we find that $\beta^* = 2\theta/(N(N-1))$, with $\theta_{ij} = \mathbb{1}\{i,j \leq d\}$ and

$$\theta = \sum_{\substack{i,j=1,\ldots,N \\ i<j}} \theta_{ij} = \frac{d(d-1)}{2}, \quad \text{and so} \quad \beta^* = \frac{d(d-1)}{N(N-1)}.$$

Then the result follows. $\qquad \square$

## G.2. General Numerical Method

At higher generality, it is difficult to make concrete statements about the behavior of $\beta^*$. Instead, we can appeal to numerical methods. Solving the optimization problem (G.3) directly is feasible using iterative solvers, but it is expensive and can be unstable. Instead, Lemma G.4 shows that $\beta^*$ is the solution to a convenient fixed point problem, and hence it can be estimated by a form of stochastic fixed point iteration.

**Lemma G.4.** *Any local minimum $\beta^*$ of $d_{\mathrm{KL}}(q_\beta \| p)$ satisfies $\beta^* = R(\beta^*)$, where*

$$R(\beta) := \frac{\mathrm{Cov}_{q_\beta}(\log w(\lambda_1, \ldots, \lambda_N), \sum_{i,j=1,i<j}^N \log |\lambda_i - \lambda_j|)}{\mathrm{Var}_{q_\beta} \sum_{i,j=1,i<j}^N \log |\lambda_i - \lambda_j|}. \tag{G.4}$$

*Proof.* Letting $Z_p$ denote the normalizing constant for $p$,

$$d_{\mathrm{KL}}(q_\beta \| p) = \log Z_p - F(\beta) - \mathbb{E}_{q_\beta} \log w + \beta F'(\beta).$$

Any local minimum of $d_{\mathrm{KL}}(q_\beta \| p)$ is also a critical point; taking derivatives in $\beta$ yields

$$\frac{\mathrm{d}}{\mathrm{d}\beta} d_{\mathrm{KL}}(q_\beta \| p) = \beta F''(\beta) - \mathbb{E}_{q_\beta} \left[ \log w \frac{\mathrm{d}}{\mathrm{d}\beta} \log q_\beta \right].$$

However, since

$$\frac{\mathrm{d}}{\mathrm{d}\beta} \log q_\beta = \sum_{i<j} \log |\lambda_i - \lambda_j| - F'(\beta),$$

which is zero-mean under $q_\beta$, it follows that any local minimum of $d_{\mathrm{KL}}(q_\beta \| p)$ satisfies

$$\beta F''(\beta) = \mathrm{Cov}_{q_\beta} \left( \log w, \sum_{i<j} \log |\lambda_i - \lambda_j| \right).$$

The result follows. $\qquad\square$

Suppose that for each $k = 1, \ldots, p$ and fixed $\beta \geq 0$, $(\lambda_1^{(\beta,k)}, \ldots, \lambda_N^{(\beta,k)}) \overset{\mathrm{iid}}{\sim} q_\beta$. Letting

$$x_{\beta,k} = \sum_{i,j=1,i<j}^N \log |\lambda_i^{(\beta,k)} - \lambda_j^{(\beta,k)}|$$

$$y_{\beta,k} = \log w(\lambda_1^{(\beta,k)}, \ldots, \lambda_N^{(\beta,k)}),$$

we can see that a consistent estimator of (G.4) is given by

$$\hat{\beta} = \frac{\sum_{k=1}^p (x_{\beta,k} - \bar{x}_\beta)(y_{\beta,k} - \bar{y}_\beta)}{\sum_{k=1}^p (x_{\beta,k} - \bar{x}_\beta)^2}, \tag{G.5}$$

where $\bar{x}_\beta = \sum_{k=1}^p x_{\beta,k}$ and $\bar{y}_\beta = \sum_{k=1}^p y_{\beta,k}$. Since (G.5) is exactly the estimator for the slope in simple linear regression, it is also unbiased. Starting from $\hat{\beta}_0 = \frac{1}{N}$, for fixed $0 < \gamma \leq 1$, consider the iterations

$$\hat{\beta}_{r+1} = \left( 1 - \frac{\gamma}{r+1} \right) \hat{\beta}_r + \frac{\gamma}{r+1} \frac{\sum_{k=1}^p (x_{\hat{\beta}_r,k} - \bar{x}_{\hat{\beta}_r})(y_{\hat{\beta}_r,k} - \bar{y}_{\hat{\beta}_r})}{\sum_{k=1}^p (x_{\hat{\beta}_r,k} - \bar{x}_{\hat{\beta}_r})^2}. \tag{G.6}$$

The following central limit theorem for $\hat{\beta}_r$ follows from Zhang (2016, Theorem 1.1) and the unbiasedness of (G.5).

**Proposition G.5.** *Let $\beta^*$ denote a critical point of $d_{\mathrm{KL}}(q_\beta \| p)$. Then for sufficiently small $\gamma > 0$, $\sqrt{r}(\hat{\beta}_r - \beta^*)$ converges in distribution to a zero-mean normal random variable as $r \to \infty$.*

Proposition G.5 suggests that the iterations (G.6) provide a reliable means of estimating $\beta^*$. In practice, we have found that $\gamma = 1$ typically suffices. The complete pseudocode is provided in Algorithm 1.

---

**Algorithm 1** Stochastic Fixed Point Iteration for Estimating $\beta^*$

---

**input** matrix size $N \in \mathbb{N}$, weight function $w$, step size $\gamma$ (e.g. $\gamma = 1$), number of samples $p$

1: Initialize $\hat{\beta}_0$ (e.g. $\beta_0 = \frac{5}{N}$)
2: **for** $r = 0, 1, 2, \ldots$ until convergence **do**
3:     **for** $k = 1$ to $p$ **do**
4:         Sample $(\lambda_1, \ldots, \lambda_N) \sim q_{\hat{\beta}_r}$
5:         Compute:
6:             $x_k = \sum_{i,j=1, i<j}^{N} \log|\lambda_i - \lambda_j|$
7:             $y_k = \log w(\lambda_1, \ldots, \lambda_N)$
8:     **end for**
9:     Compute means:
10:       $\bar{x} = \frac{1}{p}\sum_{k=1}^{p} x_k, \quad \bar{y} = \frac{1}{p}\sum_{k=1}^{p} y_k$
11:     Compute covariance and variance:
12:       $S_{xy} = \sum_{k=1}^{p}(x_k - \bar{x})(y_k - \bar{y})$
13:       $S_{xx} = \sum_{k=1}^{p}(x_k - \bar{x})^2$
14:     Update $\hat{\beta}$:
15:       $\hat{\beta}_{r+1} = \left(1 - \frac{\gamma}{r+1}\right)\hat{\beta}_r + \frac{\gamma}{r+1}\frac{S_{xy}}{S_{xx}}$
16:     **if** convergence criteria met (e.g., $|\hat{\beta}_{r+1} - \hat{\beta}_r| <$ tolerance) **then**
17:       Return $\hat{\beta}_{r+1}$
18:     **end if**
19: **end for**

---

## G.3. Variational Approximations for Commuting Matrix Models

To complete Table 2, we now look to use Algorithm 1 with symbolic regression (Cranmer, 2023) to estimate the behavior of $\kappa^*$ with respect to $m$, $n$, and the shape parameter of the Laguerre beta-ensemble. To sample from $q_\beta$, we rely on the tridiagonal matrix model of Dumitriu & Edelman (2002) (see Algorithm 2). For eigenvalue computations, we recommend an implementation of the DSTEBZ routine in LAPACK, such as the eigvalsh_tridiagonal routine in SciPy.

---

**Algorithm 2** Generating a Sample from the Laguerre $\beta$–Ensemble

---

**input** $\beta > 0$, matrix size $N \in \mathbb{N}$, shape parameter $\alpha > 0$

1: $a = \alpha + 2 + \beta(N-1)$
2: **for** $i = 1$ to $N$ **do**
3:     Sample $d_i$ from $\chi_{a-\beta(i-1)}$
4: **end for**
5: **for** $i = 1$ to $N - 1$ **do**
6:     Sample $t_i$ from $\chi_{\beta(N-i)}$.
7: **end for**

8: Build diagonal entries
$$D_i = d_i^2 + t_i^2 \quad (i = 1, \ldots, n-1), \qquad D_n = d_n^2.$$

9: Build offdiagonal entries
$$E_i = d_i t_i \quad (i = 1, \ldots, n-1).$$

10: Compute symmetric tridiagonal matrix eigenvalues with diagonal $D$, off-diagonal $E$

---

To collect estimates of $\kappa^*$ for a variety of scenarios, we consider ten values of both $m$ and $n$ logarithmically-spaced between 3 and 100 and $\alpha = 1, 2, \ldots, 5$. In Algorithm 1, we let $p = 50$, $\gamma = 1$ and choose our tolerance in each case to be $10^{-3}/N$. We then perform symbolic regression on these estimates with respect to $m$, $n$, and $\alpha$ using PySR (Cranmer, 2023) with a maximum size of 7 terms, population size of 20, 5 iterations, addition, multiplication, and division binary operations, and logarithmic and exponential unary operations.

To verify that this process is capable of reconstructing expected symbolic relationships for $\kappa^*$, we consider the symmetric

block diagonal case in Figure G.1. The predicted relationship is $\kappa^* = (m - 1.1) \times 1.05$, close to the exact relationship $\kappa^* = (m - 1) \cdot \frac{N}{N-1}$ given in Corollary G.2.

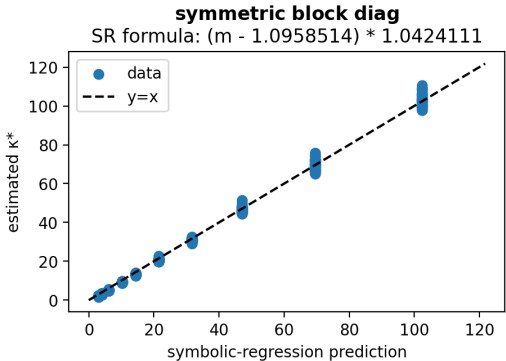

*Figure G.1.* Symbolic regression predictions vs. estimations using Algorithm 1 for $\kappa^*$ under the symmetric block diagonal structure over the Laguerre beta-ensemble.

Next, we turn our attention to the commutative cases where explicit expressions for $\kappa^*$ are not known. The results are presented in Figure G.2, which suggest the relationships:

- (b) *Commuting block diagonal.* $\kappa^* \approx \frac{1}{n}\left(m - \frac{1}{2}\right)$

- (d) *Kronecker-type.* $\kappa^* \approx \frac{n}{m} + \frac{m}{n}$.

These relationships closely follow the heuristic in Remark F.6.

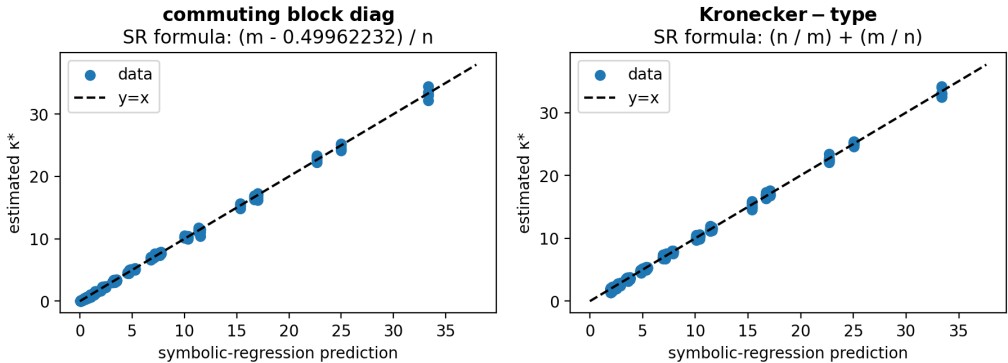

*Figure G.2.* Symbolic regression predictions vs. estimations using Algorithm 1 for $\kappa^*$ under the commuting block diagonal, and Kronecker-type matrix structures over the beta-Laguerre ensemble.

## H. Comparison of Neural Scaling Law for Linear Regression

Modern *scaling laws* in ML attempt to quantify how test errors depend on key resources such as model size and dataset size. While many such laws have been observed empirically in deep learning, there are also rigorous proofs in simpler, more tractable settings (e.g., linear models or kernel methods) that exhibit qualitatively similar phenomena. However, most previous work focuses on linear regression on the training dataset (Lin et al., 2024; Bordelon et al., 2024a).

In this section, we consider a concrete *power law* setting for the data covariance, and we review some previous results of an associated *ridge regression scaling law*, making a comparison with Proposition 5.1, which we proved in Appendix C.3. This high-dimensional ridge regression has been studied in many different scenarios, e.g., Hastie et al. (2022); Wei et al. (2022); Li et al. (2023); Defilippis et al. (2024). The main difference between our Proposition 5.1 and previous results is that we considered the ridge regression on a heavy-tailed feature matrix.

Specifically, for linear regression, we consider the dataset to be composed of, for $i = 1, \ldots, n$:

- The feature vectors $x_i \in \mathbb{R}^d$ have a covariance matrix $\Sigma$, whose eigenvalues $\{\mu_j\}_{j=1}^d$ exhibit a power-law decay

$$\mu_j(\Sigma) \sim \frac{C}{j^\alpha} \quad \text{for some constants } C > 0 \text{ and } \alpha > 0, \text{ for large } j. \tag{H.1}$$

- The label is generated via a linear model $y_i = \langle x_i, w^* \rangle + \varepsilon_i$, with i.i.d. Gaussian noise $\varepsilon_i \sim \mathcal{N}(0, \sigma^2)$.

We show how the test mean-squared error (MSE) of the ridge estimator scales with $n$, $d$, and an appropriately chosen ridge parameter $\lambda_n$. Under mild conditions (detailed below), the MSE follows a particular power law in $n$ whose exponents depend on $\alpha$ and on how we scale $\lambda_n$ with $n$. This illustrates how data spectral structure can drive nontrivial scaling phenomena in high-dimensional learning.

Given a data matrix $X \in \mathbb{R}^{n \times d}$ of $n$ samples (each row is $x_i^\top$) and labels $y = (y_1, \ldots, y_n)^\top$, the ridge regression estimator with regularization $\lambda > 0$ is defined by

$$\widehat{w} = \arg \min_{w \in \mathbb{R}^d} \left\{ \|y - Xw\|^2 + \lambda \|w\|^2 \right\} = (X^\mathsf{T} X + \lambda I_d)^{-1} X^\mathsf{T} y.$$

We will let $\lambda = \lambda_n$ possibly depend on $n$ (and/or $d$), as is common practice for controlling bias–variance tradeoffs in high dimensions (Wei et al., 2022; Li et al., 2023). The generalization error is given by

$$\mathcal{E}_{n,d}(\lambda_n) = \mathbb{E}_x \Big[ \langle x, \widehat{w} \rangle - \langle x, w^* \rangle \Big]^2, \quad \text{where } x \sim \mathcal{N}(0, \Sigma) \text{ is independent of the training set.}$$

We now present the *asymptotic behavior* of $\mathcal{E}_{n,d}(\lambda_n)$ under the spectral assumption (H.1) and a suitable scaling of $\lambda_n$ with $n$, as $n, d \to \infty$. For simplicity, let $U \Lambda U^\top$ be the spectral decomposition of $\Sigma \in \mathbb{R}^{d \times d}$, i.e., $U$ is orthonormal and $\Lambda = \text{diag}(\mu_1, \ldots, \mu_d)$ contains the eigenvalues (ordered). We can analyze ridge regression in this eigenbasis. If we let $\tilde{x}_i = U^\top x_i$, then $\tilde{x}_i \sim \mathcal{N}(0, \Lambda)$, i.e., the coordinates $(\tilde{x}_i)_j$ are independent with variance $\mu_j$. Similarly, $w^*$ can be written in the same basis, $w^* = U \tilde{w}^*$, so $w^*_{(j)} = (\tilde{w}^*)_j$ in that basis. Thus, when $x_i \sim \mathcal{N}(0, \Sigma)$, the matrix $X = [x_1^\top; \ldots; x_n^\top]$ has an SVD tightly connected with $\Sigma^{1/2}$, but we simply recall the well-known result that in the basis $U$, ridge regression amounts to a coordinate-wise shrinkage:

$$\widehat{w} = U[\widehat{\tilde{w}}], \quad \text{where} \quad \widehat{\tilde{w}}_j \approx \frac{\mu_j}{\mu_j + \lambda_n} \tilde{w}_j^* + \text{(variance term from noise)}.$$

Hence, in expectation (conditioning on $X$ or in an integrated sense), each coordinate is shrunk by a factor $\frac{\mu_j}{\mu_j + \lambda_n}$.

**Bias–variance decomposition.** For $x \sim \mathcal{N}(0, \Sigma)$, the *test MSE* can be written as

$$\mathcal{E}_{n,d}(\lambda_n) = \underbrace{\mathbb{E}\big[\|(\widehat{w} - w^*)\|_\Sigma^2\big]}_{\text{MSE in the } \Sigma\text{-inner product}} = \underbrace{\sum_{j=1}^d \mu_j \left(\text{bias}_j^2 + \text{var}_j\right)}_{\text{coordinate-wise contributions}},$$

where

$$\text{bias}_j = \mathbb{E}\big[\widehat{\tilde{w}}_j - \tilde{w}_j^*\big], \quad \text{var}_j = \mathbb{E}\Big[(\widehat{\tilde{w}}_j - \mathbb{E}[\widehat{\tilde{w}}_j])^2\Big].$$

Ignoring constants, one finds that the bias part, $\text{bias}_j \approx \big[\frac{\mu_j}{\mu_j + \lambda_n} - 1\big] \tilde{w}_j^*$, and the variance part, $\text{var}_j$, scales like $\frac{\mu_j^2}{(\mu_j + \lambda_n)^2} \cdot \frac{\sigma^2}{n}$, reflecting how the noise $\varepsilon$ passes through the resolvent $(X^\top X + \lambda_n I)^{-1}$.

As $n, d \to \infty$, we replace the sum by an integral and use that $\mu_j \sim C/j^\alpha$. Then, the test MSE is roughly:

$$\mathcal{E}_{n,d}(\lambda_n) \sim \underbrace{\sum_{j=1}^d \mu_j \left(\frac{\mu_j}{\mu_j + \lambda_n} - 1\right)^2 (\tilde{w}_j^*)^2}_{\text{bias term}} + \underbrace{\sum_{j=1}^d \mu_j \frac{\mu_j^2}{(\mu_j + \lambda_n)^2} \frac{\sigma^2}{n}}_{\text{variance term}}.$$

If $\|w^*\|^2 < \infty$ in that basis, we can assume $(\tilde{w}_j^*)^2 = \mathcal{O}(j^{-\beta'})$ for some $\beta' > 0$ or at least that $w^*$ is well-defined in $\Sigma$'s eigenbasis. For simplicity, one often takes $(\tilde{w}_j^*)^2 = \mathcal{O}(1)$ or $\mathcal{O}(j^{-\gamma'})$; we will highlight only the main effect of the *covariance* spectrum $\{\mu_j\}$ here.

**Choice of $\lambda_n$ and the Emergent Power-Law Rate.** When $\mu_j = C j^{-\alpha}$, the sums above partition into *low-index* terms (where $j$ is small and $\mu_j$ is large) and *high-index* terms (where $j$ is large and $\mu_j$ is small). Roughly speaking:

- If $\lambda_n$ is *too large*, then even the large eigenvalues $\mu_j$ get shrunk heavily, leading to big bias.

- If $\lambda_n$ is *too small*, then the small eigenvalues (at large $j$) produce a large variance.

Hence the optimal ridge parameter $\lambda_n$ in the presence of power-law $\{\mu_j\}$ often takes the form $\lambda_n \asymp n^{-\frac{\alpha}{2\alpha+1}}$, giving a trade-off that yields a final MSE scaling like

$$\mathcal{E}_{n,d}(\lambda_n^{\text{opt}}) \ \asymp \ n^{-\frac{2\alpha}{2\alpha+1}}. \tag{H.2}$$

For more details, we refer to Li et al. (2023).

**Comparison with Proposition 5.1.** The scaling limit we get in Proposition 5.1 is the product of the power of heavy tail and the ridge scaling, whereas in (H.2), the decay rate will be dominant, by the power law of the data covariance, when we choose the ridge parameter $\lambda_n$ based on $\alpha$ defined in (H.1). In this sense, our scaling limit is more general, and it really depends on the combination of the regression model structure $\lambda_n$ and the heavy tail distribution of the feature matrix. On the other hand, the above derivation of (H.2) for linear regression considered the anisotropic dataset and label noise, which is not handled in our Proposition 5.1. We leave this for future work, where we will be able to combine the heavy tail distribution in the feature matrix, the power law for the dataset covariance, and the model structure (for instance $\lambda_n$ in ridge regression). This extension may require deeper RMT for inverse Wishart matrices, such as local law results by Wei et al. (2022), or deterministic equivalence for (C.7), as in Defilippis et al. (2024).

