# OpenReview forum: "Models of Heavy-Tailed Mechanistic Universality"
_ICML.cc/2025/Conference — ICML 2025 poster_

### Official Review · Reviewer_PqrH · 2025-03-12

**Overall Recommendation:** 3

**Summary:**

Recent advancements in deep learning, including neural scaling laws, have highlighted the prevalence of heavy-tailed or power law behaviors in key network components such as the Jacobian, Hessian, and weight matrices. This phenomenon, termed heavy-tailed mechanistic universality (HT-MU), has been empirically linked to model performance, suggesting its fundamental role in deep learning success. To investigate the origins of this behavior, the study introduces a general class of random feature matrix models, the high-temperature inverse-Wishart ensemble. The model identifies three key factors contributing to heavy-tailed spectral densities: (i) complex correlation structures in data, (ii) lower training temperatures, and (iii) implicit bias in model structure leading to reduced eigenvector entropy. The study further explores the implications of HT-MU on learning dynamics, neural scaling laws, and optimizer trajectories.

**Claims And Evidence:**

Yes, the claims made in the submission are supported by clear and convincing evidence.

**Essential References Not Discussed:**

No, there are not related works that are essential to understanding the (context for) key contributions of the paper, but are not currently cited/discussed in the paper.

**Experimental Designs Or Analyses:**

Yes, I checked the soundness/validity and experimental designs or analyses. The numerical experiments of the paper effectively validates the results of its theoretical analysis.

**Methods And Evaluation Criteria:**

No, this paper doesn't propose any method or evaluation criteria.

**Other Comments Or Suggestions:**

See weaknesses.

**Other Strengths And Weaknesses:**

**Strengths:**
1. This work proposes a novel high-temperature inverse-Wishart ensemble model, providing a unified theoretical framework to explain the emergence of the power-law phenomenon in deep learning.

2. The paper presents comprehensive numerical experiments, effectively empirically validating the effectiveness of its theory.

3. Overall, the paper is well-written and easy to follow.

**Weaknesses:**
1. The overall writing density of the paper is somewhat high. It is recommended to move some less important discussions on statistical physics to the appendix.

2. It is suggested that the authors add a separate experiment section in the main text to provide details on the experimental setup, conclusions, and their analysis.

3. Regarding the theoretical analysis of the neural scaling law, the authors seem to have only derived the power-law relationship concerning the amount of training data, without considering factors such as model size. It is recommended that the authors provide a more detailed discussion on the relationship between the high-temperature inverse-Wishart ensemble model and the neural scaling law.

**Questions For Authors:**

See weaknesses.

**Relation To Broader Scientific Literature:**

To uniformly analyze and understand the power-law phenomenon in deep learning, the paper proposes the high-temperature inverse-Wishart ensemble model.

**Theoretical Claims:**

Yes, I checked the correctness of the proofs for theoretical claims. Proposition 3.1 provides the density of 'optimal features,' which suggests that a stochastic optimizer concentrates on regions with high marginal likelihood (also called model evidence).

---

> ### Author Rebuttal · Authors · 2025-04-01
>
> We thank the reviewer for their positive assessment of our work and for taking the time to verify the correctness of our proofs. We appreciate that the reviewer finds the experiments to be sound and suitably validate our model class. We agree about the high writing density; there are many individual components and motivations in our discussion. In line with other reviewer comments, we are considering alterations to the second section to better emphasize relevant details and move away from some of the statistical physics (see comments to Reviewer RfHj for example). We will include all experimental details in the main text by adding a separate experiment section with the following:
>
> 5+1 phases of learning experimental details: For Figure 5, we train a MiniAlexNet for the CIFAR10 classification task with different batch sizes. This MiniAlexNet is a simplified version of AlexNet, which contains six layers: the first three are convolutional layers, followed by max-pooling layers, and the last three are fully connected layers. The histograms in Figure 5 are the histograms of the eigenvalues of $WW^T$ where $W$ is the trained weight matrix in the first fully-connected layer with input dimension $192 * 4 * 4$ and output dimension 1000. Here $W$ is initialized by the centered normal distribution with variance $\sqrt{1/\mathrm{fan_in}}$. For all the histograms in Figure 5, we trained the network using SGD with momentum, with a learning rate of 0.01 and a momentum parameter of
> 0.9 for 200 epochs. For each histogram, we repeat the experiments 3 times for the average. The red dot curves are the numerical simulations of the density function of the HTMP with different $\kappa$ and $\gamma=1000/(192 * 4 * 4)$.
>
> To obtain NTK spectral densities, we consider larger neural networks that are trained to near-zero loss (all $>99.8\%$) on a subsampled dataset of 1000 entries through 200 epochs of a cosine annealing learning rate schedule with 200 epoch period, starting from a learning rate of 0.05 with a batch size of 64. Each model is comprised of the following number of parameters:
> - resnet9 (4.8M parameters)
> - resnet18 (11.1M parameters)
> - vgg11 (9.2M parameters)
> - vgg13 (9.4M parameters)
> - lenet (62K parameters)
> - logistic (30K parameters)
> - densenet121 (7.0M parameters)
> The output layer of each model is altered from their ImageNet counterparts to classify with ten classes (for CIFAR-10, SVHN, and MNIST datasets).
>
> We also recognize the relationship of the amount of training data to be a limitation. It is possible to complete the scaling law by including the power law dependence on the number of model features, although this differs from model size in a strict sense. Unfortunately, without a precise parameterization, it is difficult to establish such a law, but along with dependencies on individual model properties (e.g. depth, see comments to Reviewer Weri), we consider this of prominent interest in follow-up work.

---

### Official Review · Reviewer_Weri · 2025-03-12

**Overall Recommendation:** 5

**Summary:**

This paper argues that many phenomena observed in neural scaling laws arises from universal random matrix theory effects which the authors term heavy tailed universality. The paper introduces a theory which breaks up the deep network optimization into an optimization over features and optimization over the last layer weights. The relative strength of learning for these different components are controlled by by a hyperparameter $\rho$. This leads to a *master model* for the feature matrices that the authors are able to compute the eigenvalue densities of such adapted kernels using techniques from random matrix theory. Depending on the hyperparameters of the model, there can be 6 types of observed feature spectra. The authors make connections to scaling law literature.

**Claims And Evidence:**

The theoretical claims are supported by both proofs and numerical experiments.

**Essential References Not Discussed:**

There are several important references that were not referenced but I think should possibly be included

1. Li et al analyze how the hidden layer kernels in a Bayesian neural network change in a proportional limit where width and data diverge at fixed ratio https://journals.aps.org/prx/abstract/10.1103/PhysRevX.11.031059.
2. Zavatone Veth et al also analyze the distribution over kernels after training in Bayesian linear networks in a variety of scaling limits https://ieeexplore.ieee.org/abstract/document/9723137
3.  Thamm et al empirically study the spectra of trained weight matrices in deep networks https://arxiv.org/abs/2203.14661. Their results may also be relevant to the final feature matrices.
4. Bordelon et al 2024 studied the dynamics of a random feature model concurrent with the work of Paquette et al 2024 and Lin et al 2024, also capturing compute optimal laws while also treating the effect of limited data in addition to limited training time or features https://openreview.net/pdf?id=nbOY1OmtRc.  A recent follow up from that group rederived their result using techniques from random matrix theory https://arxiv.org/abs/2502.05074 including $S$-transform methods.
5. Bordelon et al 2024 had a follow up where they considered a simple model of rich (non-kernel) learning dynamics where the kernel is allowed to adapt that reproduced a faster scaling law in terms of source and capacity exponents https://arxiv.org/abs/2409.17858. This can be attributed to the changes to the kernel during optimization. I sense there could be a connection between their results and the theory in this work which allows the feature kernel to adapt beyond its prior.

**Experimental Designs Or Analyses:**

Yes, the experiments appear correct.

**Methods And Evaluation Criteria:**

The authors focus on smaller scale vision models but this is very appropriate for a theoretical paper.

**Other Comments Or Suggestions:**

1. It could be useful to the reader to quickly define or outline each of the 5+1 phases either in the main text or in the supplementary material so that the reader would not need to consult prior works.
2. In line 951 it says "Appendix _"

**Other Strengths And Weaknesses:**

The paper is technically strong and provides many experiments to support its claims. There are some remaining questions about how architectural details like depth and other earlier layers alter their master model (see questions below).

**Questions For Authors:**

1. What is the role of model depth and other architectural details in this model (like widths of earlier layers)? Does all of this enter into the prior distribution over the feature matrix $\pi(M)$? Would it also effect the rate at which the feature matrix can evolve during learning? What if an earlier layer had a significant bottleneck in width compared to the final feature layer. This would likely change the expressivity of the network and decrease the flexibility of the final feature matrix during optimization.
2. In section 5.3, the authors consider optimization trajectories. However, the theory provides the distribution for the *final features.* Could the authors comment on the connection? Are they assuming that the features equilibrate more rapidly than the readout and thus the loss?
3. The authors claim that other works rely on power law assumptions in the data space rather than feature space. This is not universally true, see for example section 5.1 here https://arxiv.org/abs/2409.17858 where the data is uniformly distributed but the nonlinearity in the network causes a power law decay which sets the scaling law on that task. Could the authors revise this?

**Relation To Broader Scientific Literature:**

This paper studies an important problem of the origin of neural scaling laws and provides an interesting universality hypothesis about their origin beyond the lazy learning regime.

**Theoretical Claims:**

Yes, I read through many of the derivations in the Appendix.

---

> ### Author Rebuttal · Authors · 2025-04-01
>
> We thank the reviewer for their positive assessment of our work, and for providing several references whose inclusion greatly enhances the working document! In particular:
>
> - Thamm et al. do an excellent job of highlighting the phenomenon we are interested in and provides further evidence for our case (their Figure 4 shows a proportion of eigenvectors are non-uniform)
> - We thank the reviewer for providing the neural scaling law paper by Bordelon et al. 2024, and we admit that this paper considered the power law assumption for the Fourier spectra of the kernels at initialization. This feature learning setup is similar to our power law assumption of the feature map and is the motivation of our neural scaling law section. They also consider a similar setup to the simultaneous training of weights and features in our Proposition 3.1. Very cool! We will modify the literature review in our neural scaling law section.
>
> The other references seem to provide excellent examples of more fine-grained analyses with other precise models of training and architectures. We agree these provide excellent context and a better illustration of the state-of-the-art in the theory of feature learning.
>
> In addition, we include the following references:
> - (Pillaud-Vivien et al., 2018) also assume power law features.
> - (Mezard & Montanari, 2009) for statistical mechanics of learning
> - (Yang et al., 2023) "Test Accuracy vs. Generalization Gap" instead of Martin & Mahoney, 2021a for review papers on robust metrics
>
> We recognize that we did not summarize the 5+1 phases of learning. We include the following at line 363:
>
> In a sequence of papers, Martin & Mahoney observe 6 classes of empirical behaviors in trained weight matrices, comprising a smooth transition from a random-like Marchenko-Pastur to a heavy-tailed density, before experiencing rank collapse. Excluding rank collapse, the five primary phases are:
> (a) Random-Like: Pure noise, modeled by a Marchenko-Pastur density.
> (b) Bleeding-Out: Some spikes occurring outside the bulk of density.
> (c) Bulk+Spikes: Spikes are distinct and separate from the Marchenko-Pastur bulk.
> (d) Bulk-Decay: Tails extend so that the support of the density is no longer finite.
> (e) Heavy-Tailed: The tails become more heavy-tailed, exhibiting the behavior of a (possibly truncated) power law.
> The transition from (a) to (e) is also seen in (Thamm et al., 2022). This smooth transition between multiple phases is a primary motivation of this work. We find that this behavior is displayed by a combination of a nontrivial covariance matrix to capture the spikes, and the HTMP class with decreasing $\kappa$.
>
> To answer your questions:
> 1. The depth and architectural effects of the structures are challenging to analyze due to the adopted general approach. These aspects can be studied empirically, necessitating further research. We admit that the model depth and other architectural details in this model will strongly affect the prior distribution of the feature matrix $\Phi$ and also the final feature matrix. In this work, we want to provide a new random matrix model (high-temperature MP law) which may resemble the spectral properties of the feature matrix $\Phi$. By tuning the parameters in HTMP, we can mimic the spectra of the feature matrices in different scenarios, which may be at initialization, may have been well trained, or may be associated with very complicated architectures. Although we do not have a clear picture of the relationship between the architectural details and spectra of the feature matrices, our random matrix model can potentially be viewed as an equivalent simplified model to study the feature matrices with very different architectural details.
> 2. Yes, we are assuming that features equilibrate more rapidly than the loss. In this way, we are examining trajectories at the end of training (the kernel learning regime in (Fort et al., 2020) "Deep learning vs. kernel learning") once the features are mostly trained. We elaborate more on this in the current version.
> 3. Interesting! So the input data is uniform, but this example still seems to assume a power law Fourier spectrum in the target function. Unless we are missing something, this still assumes heavy tails in the data (in the labels in this case). However, we have found that (Liao & Mahoney, 2021) "Hessian Eigenspectra" also shows how nonlinearities can influence the spectrum, although they do not prove scaling laws from this. In this case, conditions for the nonlinearity to exhibit heavy tails are still unclear, so we consider this a case of examining individual models.

---

> > ### Comment · Reviewer_Weri · 2025-04-02
> >
> > I appreciate the authors' thoughtful response. I think with the improved detail about the 5 phases and the improved comparison to prior works, this paper provides an novel and useful theoretical framework to model feature adaptation from the perspective of random matrix theory. Future works could extend this framework to incorporate more architectural details such as depth, nonlinearity, etc. In light of this, I will increase my score as I strongly favor acceptance.

---

### Official Review · Reviewer_RfHj · 2025-03-13

**Overall Recommendation:** 1

**Summary:**

This paper explores heavy-tailed mechanistic universality (HT-MU) in deep learning by proposing a new family of random feature matrix models based on the "high-temperature inverse-Wishart" ensemble. The paper reviews two mechanisms and presents a third one for the emergence of power laws in different matrices related to trained neural networks. These three mechanisms are (i) complex correlations in data, (ii) reduced training temperatures, and (iii) implicit architectural biases that affect eigenvector entropy. The authors provide theoretical results linking their model to neural scaling laws, the five phases of learning, and optimizer trajectory properties.

**Claims And Evidence:**

While the theoretical framework is ambitious, the logical flow of the paper is extremely hard to follow.
The paper seems to review and reject the PIPO (Population Covariance) approach; they reject it based on "More recent analyses have shown how architectural decisions can alter the power law, but these hold only for specialized models." (lines 245), however, they give no reference to such recent analysis.
Recursive model structure is reviewed and rejected in much the same manner.
Finally, they come to their suggestion of reduced eigenvector entropy. In this section, the manuscript makes a number of confusing statements.
- What do the authors mean by assuming $\Phi$ describes a positive-definite matrix? $\Phi$ was introduced as $n \times m$ feature mapping, with $n$ being the number of datapoints and $m$ the number of features. Generally this is not a square matrix, nor is it positive definite. When is such an assumption expected to hold?
- The manuscript moves on to stating "In this change of variables, it is typically assumed that the distribution of eigenvectors (Q) is uniform." Uniform over what domain? (the d-sphere?) Why is this assumption typical? It would be helpful to cite relevant works.
- The concept of eigenvector entropy is never introduced and a reference for it is never given. In fact, I was hard pressed to find references to it in the literature.
- The paper goes on to talk about "free eigenvectors", these are never defined.
- This is followed by: "Admittedly, the link between (5) and non-uniform eigenvector distributions is nontrivial, so several points of evidence are in order",  the status of much of the statements that follow is unclear, is it a conjecture, an interpretation, or a known result?


The paper follows with applications. They start with the 5+1 phases of learning, this phenomenon is not nearly as widely recognized as scaling laws, for the manuscript to be self-contained, it should include at least a brief explanation of what these are.

The authors need to overhaul the presentation to clearly delineate how their evidence supports each claim, ideally with concrete examples or references.

**Essential References Not Discussed:**

The whole choice on references should be revised.

**Experimental Designs Or Analyses:**

The authors show that finite temperature inverse Wishart distributions fit experimental data in several settings which they label according to a nomenclature which is not clarified in the text. Putting the latter point aside, since the theoretical basis for this distribution isn't clear, much further numerical evidence and precision tests are needed in our minds to make this claim sound.

**Methods And Evaluation Criteria:**

See above

**Other Comments Or Suggestions:**

In Table 1, the manuscript mentions "observations" but gives no reference to these observations.

The manuscript's notation is confusing and inconsistent.

1. Eq. 3 includes $\tau$. $\tau$ is not presented before in the manuscript and it is only presented in Appendix G.1.
2. $\rho$ has at least three meanings, two appear in the "metatheorem", and then $\rho$ is introduced again as the ratio $\gamma / \eta$.
3. What is the definition of $N$ in section 4.3 and on? In section 5.2. it seems to be the dataset size, which was previously denoted by $n$.

**Other Strengths And Weaknesses:**

Strengths

The paper takes on an ambitious goal of establishing and explaining a universal behavior in deep neural networks. This goal is ambitious and inspiring.

Weaknesses

The paper should be rewritten. At the current state, I gauge it would be inaccessible to the large majority of the ICML community, and the large majority of those whose research field is theory of DL.

**Questions For Authors:**

Questions:
1. Below Eq.3 is it assumed that $M=\Phi \Phi^T$ is invertible? Is that mentioned in the manuscript? Is that supposed to be a pseudo-inverse?
2. Could the authors specify what they mean by "For more general classes of models, computing marginal likelihood becomes intractable" in line 195? What is being generalized? Is it the choice of loss function?
3. Later in line 207 the manuscript focuses on the scenario "However, if L(Θ∗, Φ) is constant in Φ , that is, any choice of Φ yields the same training loss". What could be an example of such a scenario?
4. Under the "Activation Matrices" subsection $y$ is introduced as a regression target, but in Appendix $G.1$ $y$ has a different definition, together with a new notation $\epsilon$ which is never introduced. Could the authors clarify their setting? Do the two definitions of $y$ somehow coincide?

**Relation To Broader Scientific Literature:**

The manuscript cites works of Mahoney extensively, in a way that stands out compared to usual academic standards. This choice stands in sharp contrast to the relative thin reference to existing literature, e.g., citing the work of Martin & Mahoney along side a single other paper as examples of "statistical mechanics of learning", though it is not a review or a book, and there are much more prominent works in the field. The same repeats for "robust metrics to assess model quality", a work by Martin & Mahoney is cited together with a single additional paper, again, for such a rich field, a review would be more useful to the reader, or a choice of more canonical works.
At the same time, the paper simply lacks crucial references, for example, the paper mentions the Donsker-Varadhan variational formula (Appendix G.1) but does not cite the paper.

Additionally, the paper is full of jargon that is not common in the literature, for example, the paper cites Arous & Guionnet, 2008 for the statement "While independent matrix elements exhibiting near-ballistic power laws can give rise to heavy-tailed spectral densities" but the jargon "ballistic" does not appear in the cited work.
To continue this problematic line of misreference, they cite Hanin & Nica, 2020 to support the statement "These results assert power laws in the elements of feature matrices." but Hanin & Nica do not mention power laws.

These are just some examples I checked, I'm sure they are not the only ones.

I recommend the authors completely revise their choice of references, as they currently give a partial and skewed view of the literature.

**Theoretical Claims:**

The paper makes it difficult to assess its claims. It is often hard to distinguish between known results, conjectures, and rigorous results in the text.

---

> ### Author Rebuttal · Authors · 2025-04-01
>
> We thank the reviewer for their careful examination of our work and for the thoughtful feedback. We appreciate that they find our framework to be ambitious, and understand the concern about empirical verification of individual claims within our theory. One of our fundamental assumptions---that the eigenvectors are not uniformly distributed---was verified by a reference provided by Reviewer Weri; (Thamm et al., 2022), and we will include the reference to this. However, our primary goal in this paper is the development of a single theoretical model designed explicitly to display observed asymptotic spectral tail behavior (Theorem 4.3). Our discussion is used to motivate the construction of a variational family with the desired characteristics. We study this model to compare its overall predicted tail behavior against empirical phenomena. We cannot claim that this model represents specific architectures or matches empirical phenomena identically; our analysis cannot replace other fantastic work that is being conducted in this direction. To help focus our claims, we add the following to line 53:
>
> Our central objective is to identify a plausible random matrix class that exhibits inverse Gamma law spectral behavior, and the smooth transition from Marchenko-Pastur to heavier-tailed densities observed in Martin & Mahoney (2021) and Thamm et al. (2022).
>
> and replacing 63--65:
>
> - construct a parametric family of spectral densities which includes the Marchenko-Pastur law, but allows for heavy-tailed spectral tail behavior in line with empirical observations (Figures 1 \& 5, Appendix I)
>
> We replace lines 125--135 with the following:
>
> However, PIPO also fails as a standalone learning theory, as it does not account for implicit model biases. If data _alone_ influences tail behavior, models trained on the same dataset should exhibit similar power laws (Section 4.1), which contradicts empirical findings (Yang et al., 2023). At present, theoreticians analytically examine the interactions of individual models in the presence of heavy-tailed data (Maloney et al., 2022), but such analyses are intractable at scale. Originally conjectured by Martin & Mahoney (2021), here, we look for a third alternative: a universal, model-agnostic mechanism that can give rise to different heavy-tailed spectral behaviors from the same dataset. We construct a sequence of hypotheses that lead to such a mechanism, which we refer to as "eigenvector entropy". Matrix models in the literature typically have eigenvectors that are Haar-uniformly distributed (maximum entropy) (Anderson et al., 2010); this is also represented by _delocalization_ (Bloemendal et al. 2014). Breaking this property, reducing entropy, our framework provides a family of variational approximations (HTMP) with the right qualitative behavior: arbitrary power law spectra, and inverse Gamma laws that can arise from model design alone.
>
> Regarding more specific comments:
> - $M$ should be positive-definite, not $\Phi$ (typo)
> - "Near ballistic" has been changed to "power law exponents $\alpha < 3"
> - Donsker-Varadhan now cited
> - Hanin & Nica discuss heavy-tailed log-normals; not power laws, but can appear indistinguishable, see (Clauset et al., 2007). We now make this clear.
> - "Free eigenvectors" replaced with "eigenvectors $v_1,...,v_N$ with $v_1,...,v_d$ Haar-uniform and $v_{d+1},...,v_N$ fixed.
> - The approximation (6) is proposed in accordance with patterned matrix models whose spectral densities we calculate in Appendix D. It is not exact, but designed to replicate qualitative phenomena.
> - See response to Reviewer Weri regarding 5+1 phases.
> - For further citations to cover claims made about the wider literature, see the response to Reviewer Weri.
> - Section 2.5 of (Anderson et al. 2010. An Introduction to Random Matrices.) showed that eigenvectors of GOE/GUE are Haar uniformly distributed; we now include this.
> - For general Wigner or sample covariance matrices, (Bloemendal et al. 2014. Isotropic local laws for sample covariance and generalized Wigner matrices) used local law results to prove the delocalization for all eigenvectors, which means all the entries of the normalized eigenvectors are $1/\sqrt{N}$ order. For more details, we refer to local law lecture note by (Benaych-Georges \& Knowles 2016).
>
> To answer your questions:
> 1. Yes, but this assumption is not needed if $M = \Phi \Phi^\top + \frac{\gamma}{2\tau} I$, so we take this in the updated version.
> 2. Thank you for the suggestion; this now reads "For more general losses..."
> 3. Linear regression with equal numbers of parameters and datapoints is one example, since $\min_\Theta L = 0$.
> 4. The Appendix includes $L^2$ regularization, but otherwise the two are equivalent. We now add $L^2$ regularization to the activation matrices section.
>
> Please let us know if you have remaining questions about claims; space constraints prevent us from addressing further comments, but we are happy to provide evidence in the discussion period.

---

> > ### Comment · Reviewer_RfHj · 2025-04-08
> >
> > I thank the authors for the clarifications and responses. The answers and corrections to the manuscript are certainly an improvement. However, I believe a major revision is required to make the paper accessible and accurate. I thus recommend rejection, and I encourage the authors to submit a majorly revised version to the next round.

---

### Decision · Program_Chairs · 2025-05-01

**Decision:**

Accept (poster)

**Comment:**

The submission focuses on the origins of power-law spectra in deep learning by introducing a new framework based on the so-called high-temperature inverse-Wishart ensemble as a model for random features. The authors classify the possible sources of heavy-tailed spectra and characterise the resulting spectral properties within the proposed model. The manuscript is found to be technically strong and possibly an advancement in the study of the relevance heavy tailed feature matrices in machine learning. Although the received feedback is overall positive, reviewers complained about its dense mathematical content and not clear expostion. I strongly recommend the authors to tackle a review of the text by fixing the typo and address the concerns of Reviewer ```RfHj``` about the bibliography and the large number of citations of works by Mahoney and coworkers by considering their relevance for the text. I suggest *Weak accept*.